# Generalization Performance Gap Analysis between Centralized and Federated Learning: How to Bridge this Gap?

## Abstract

The rising interest in decentralized data and privacy protection has led to the emergence of Federated Learning. Many studies have compared federated training with classical training approaches using centralized data and found from experiments that models trained in a federated setup with equal resources perform poorly on tasks. However, these studies have generally been empirical and have not explored the performance gap further from a theoretical perspective. The lack of theoretical understanding prevents figuring out whether federated algorithms are necessarily inferior to centralized algorithms in performance and how large this gap is according to the training settings. Also, it hinders identifying valid ways to close this performance distance. This paper fills this theoretical gap by formulating federated training as an SGD (Stochastic Gradient Descent) optimization problem over decentralized data and defining the performance gap within the PAC-Bayes (Probably Approximately Correct Bayesian) framework. Through theoretical analysis, we derive non-vacuous bounds on this performance gap, revealing that the difference in generalization performance necessarily exists when training resources are equal for both training setups and that variations in the training parameters affect the gap. Moreover, we also prove that the complete elimination of the performance gap is only possible by introducing new clients or adding new data to existing clients. Advantages in other training resources are not feasible for closing the gap, such as giving larger models or more communication rounds to federated scenarios. Our theoretical findings are validated by extensive experimental results from different model architectures and datasets.

## 1 Introduction

Classical deep learning algorithms are typically performed in centralized settings (LeCun et al., 1998; He et al., 2016; Vaswani et al., 2017). Specifically, deep neural networks are trained with massive amounts of data on servers equipped with strong computation power. Enormous research and projects have proven this training setup to work well. For example, Large Language Models (LLMs) (Brown et al., 2020; Lieber et al., 2021; Black et al., 2022; Hoffmann et al., 2022; Thoppilan et al., 2022), which have recently received significant attention due to their impressive performance on various tasks, are generally trained with the centralized setup. However, an inherent limitation of this approach is the imperative centralization of training data (Chen et al., 2023). In reality, the majority of data is generated and stored in a distributed manner. If data containing sensitive information is centralized, the privacy of participating parties will likely be compromised. The challenge of expanding data size while protecting data privacy has led to the emergence of a new type of learning methods that exploit training with distributed data. One such popular method is called Federated Learning (McMahan et al., 2017; Zhuang et al., 2021; Karimireddy et al., 2020; 2021; Tang et al., 2022). In the training scenario of Federated learning, the training data is preserved on participating clients, and multiple clients collaborate with a central server to train a model without sharing data (Li et al., 2021).

The introduction of federated training effectively alleviates the privacy problem, but there is no perfect solution (AbdulRahman et al., 2020). By comparing the two types of training setups, many studies have found that under equal training resources, the models trained in a federated scenario

do not perform as well as models trained in a centralized scenario in test datasets or downstream tasks (Elnakib et al., 2023; Zhao et al., 2018), which drastically hinders the broad application of federated learning. Notably, this conclusion was established through empirical evidence, and the theoretical aspect has yet to be fully explored (Garst et al., 2023; Mar'i et al., 2023). The lack of theoretical understanding has resulted in long-term arguments on the existence of the performance gap (Drainakis et al., 2023). Also, it prevents the identification of appropriate ways to close this gap. Significant resources have been wasted on repeated experiments in search of promising directions.

In this paper, we re-visit the question: *Given the same model, training data, and total training compute, can federated learning catch up with or surpass centralized learning in terms of generalization performance?* To advance the theoretical underpinnings, we model two types of learning as Stochastic Gradient Descent (SGD) (eon Bottou, 1998; Sutskever et al., 2013) optimization problems on centralized and decentralized data, respectively, and establish PAC-Bayes (Probably Approximately Correct Bayesian) bounds (McAllester, 1998; 1999) on the generalization error of the models trained in each training setup. Since the generalization bound is usually considered an essential index of the generalization ability of the learning algorithm, the performance gap is formulated as the distance between two generalization bounds. By analyzing this distance equation, we find that the number of clients positively correlates with the performance gap and derive non-vacuous lower and upper bounds on the performance gap. These bounds theoretically show that the performance gap will necessarily exist under equivalent training conditions and is affected by the training settings. Therefore, completely bridging the performance gap requires federated scenarios to be provided with more training resources. Following this idea, we theoretically prove that the complete close of the performance gap is only possible by incorporating new clients or adding data to existing clients. In addition to theoretical analyses, we also empirically explore the performance gap by conducting extensive experiments. To ensure our theoretical findings can be generalized to different models and data, we chose two popular architectures, ResNet (He et al., 2016) and Vision Transformer (Dosovitskiy et al., 2020), and collected their training and testing results on two standard datasets, CIFAR-10 (Krizhevsky et al., 2009) and Mini-ImageNet (Vinyals et al., 2016; Deng et al., 2009). The experimental data is found to be closely aligned with our theoretical conclusions.

In summary, the key contributions of our paper are shown below:

1. We introduce a novel theoretical perspective to understand the performance gap between centralized and federated training, defining this gap as the distance between the PAC-Bayes generalization bounds of two scenarios.

2. We prove that the performance gap monotonically increases with the number of clients and establish non-vacuous lower and upper bounds on this gap, demonstrating that the gap inevitably exists when two training scenarios are provided with equivalent training resources. Our analysis also reveals the influence of training settings on this gap.

3. We derive that only introducing new clients or adding data to existing clients are possible to completely bridge the performance gap. Other approaches, such as scaling up model size or increasing communication rounds, cannot fully close this gap.

4. Extensive experiments on different model architectures and datasets validate the correctness of our theoretical results.

The rest of this paper is structured as follows. We review some related works in Section 2. We introduce the necessary preliminaries in Section 3. We show our theoretical analyses of the performance gap in Section 4, followed by the empirical validation of our theoretical findings in Section 5. Finally, we give a conclusion of the paper in Section 6. The Appendix presents the details omitted from the main manuscript.

## 2 RELATED WORKS

### 2.1 FEDERATED LEARNING

Federated learning is a class of distributed learning methods proposed for collaborative model training without compromising privacy (AbdulRahman et al., 2020; Li et al., 2021). The benchmark algorithm for federated learning is Federated Averaging (FedAvg) (McMahan et al., 2017). This

algorithm first introduces the scenario of federated learning, consisting of massive decentralized clients and a central server that establishes communications with all clients. During training, multiple clients train models received from the server using local training data, and then the server aggregates the training updates received from these clients to update the model. Client privacy is protected as no local data is shared during training. In recent years, as people have become aware of the importance of data privacy for security, many research works related to federated learning have emerged (Zhuang et al., 2021; Tang et al., 2022; Zhao et al., 2018; Tran et al., 2019). These works generally hold the impression that centralized learning must perform better than federated learning, and many of them focus on proposing advanced federated algorithms to catch up with the centralized baseline (Karimireddy et al., 2021; Zhuang et al., 2021). However, the correctness of this impression has not been fully explored from a theoretical aspect. Our work fills this gap and identifies generic strategies that can bridge the gap between the two training setups.

## 2.2 COMPARE FEDERATED LEARNING WITH CENTRALIZED LEARNING

Since federated learning was proposed, there have been studies focusing on the comparison between federated and centralized training. Some works aim to compare the performance of the models trained in each training scenario. These comparative evaluations report that models trained in a centralized setup generally outperform models trained in a federated setup across a variety of tasks and datasets, such as MNIST (Peng et al., 2022; Mar'i et al., 2023), CIFAR-10 (Zhao et al., 2018), and CICIDS2017 (Elnakib et al., 2023). Similar experimental results are also found in the federated studies that adopt the centralized training results as one of the baselines (Zhuang et al., 2021). In addition to performance comparison, there are comparisons on the training convergence rate. Unlike the above studies, these studies show that federated algorithms can attain the same order or faster convergence rate than centralized algorithms (Karimireddy et al., 2020; 2021; Asad et al., 2021). Furthermore, a recent study by Drainakis et al. explores the differences between federated and centralized training from the perspectives of energy cost and bandwidth cost (Drainakis et al., 2023). However, most of these works have primarily offered observational insights based on empirical evidence, especially those targeting the performance gap. The lack of theoretical underpinnings has prevented researchers from explaining how the performance gap develops and proving to others whether it necessarily exists. To address this shortcoming, we quantify the performance gap as a bounded analytic solution and theoretically analyze it in this paper.

## 2.3 GENERALIZATION BOUND FOR STOCHASTIC ALGORITHMS

Stochastic Gradient Descent (SGD) (eon Bottou, 1998; Sutskever et al., 2013) is a foundational optimization method in machine learning (LeCun et al., 1998; Hinton & Salakhutdinov, 2006; Goodfellow et al., 2014; McMahan et al., 2017; Tang et al., 2022). Extensive research has quantified the generalization abilities of stochastic algorithms through PAC-Bayes upper bounds (He et al., 2019; Mou et al., 2018; London, 2017; Pensia et al., 2018) and utilized these bounds to study different aspects, including algorithm convergence (Mou et al., 2018; Pensia et al., 2018), training stability (Zhu et al., 2024), and hyper-parameter tuning strategies (He et al., 2019). The generalization bound also plays an important role in research works related to federated learning (Yuan et al., 2021). Several studies propose new training frameworks to tackle problems such as non-IID data distribution (Zhao et al., 2024; Sun et al., 2024b) and model personalization (Boroujeni et al., 2024; Achituve et al., 2021; Vedadi et al., 2024) based on this bound. Moreover, this bound has been used to understand the impact of the parameters (Sefidgaran et al., 2024) or the network structure (Sun et al., 2024a) on generalization. However, existing works focus on using the generalization bound to analyze a single learning regime, unconcerned about the difference between centralized and federated training in generalization. We establish a theoretical expression for the generalization gap according to the distance between the generalization bounds of stochastic algorithms in both settings.

## 3 PRELIMINARIES

### 3.1 GENERALIZATION ERROR

In machine learning, let the hypothesis class of a model be denoted as $\Theta \subset \mathbb{R}^d$. The primary goal of learning algorithms is to identify a parameter vector $\theta \in \Theta$ that minimizes the expected risk,

expressed as $\mathcal{R}(\theta) = \mathbb{E}_{\xi \sim \mathcal{D}} F(\theta; \xi)$. Here, $d$ represents the dimension of $\Theta$, $F$ is the loss function, and $\mathcal{D}$ is the unknown distribution of the test data. When the parameter $\theta$ is treated as a random variable following a distribution $Q$, the expected risk with respect to $Q$ can be written as:

$$\mathcal{R}(Q) = \mathbb{E}_{\theta \sim Q} \mathbb{E}_{\xi \sim \mathcal{D}} F(\theta; \xi). \tag{1}$$

Since the true data distribution $\mathcal{D}$ is typically unknown, the expected risk $\mathcal{R}$ is approximated by the empirical risk $\hat{\mathcal{R}}$, based on the training data's distribution $\hat{\mathcal{D}}$, as follows:

$$\hat{\mathcal{R}}(Q) = \mathbb{E}_{\theta \sim Q} \mathbb{E}_{\zeta \sim \hat{\mathcal{D}}} F(\theta; \zeta). \tag{2}$$

The discrepancy between the expected risk $\mathcal{R}$ and the empirical risk $\hat{\mathcal{R}}$ is what defines the generalization error.

### 3.2 PAC-BAYES UPPER BOUND FOR GENERALIZATION ERROR

Within the PAC-Bayes (Probably Approximately Correct Bayesian) framework (McAllester, 1998; 1999), hypothesis functions learned by stochastic algorithms are viewed as randomly sampled functions from a hypothesis class. The generalization ability of an algorithm is measured by the distance between the posterior distribution of the output hypothesis $Q$ and the prior distribution $P$, which is typically assumed to be Gaussian or Uniform. This leads to a classic result that provides a uniform bound on the expected risk $\mathcal{R}(Q)$, presented as follows:

**Lemma 1.** *For any positive real number $\delta \in (0, 1)$, and for all distributions $Q$, the following inequality holds with probability at least $1 - \delta$ over a sample of size $N$:*

$$R(Q) \leq \hat{R}(Q) + \sqrt{\frac{\mathcal{D}(Q||P) + \log(\frac{1}{\delta}) + \log(N) + 2}{2N - 1}}. \tag{3}$$

*where $\mathcal{D}(Q||P)$ denotes the KL divergence between $Q$ and $P$, defined as:*

$$\mathcal{D}(Q||P) = \mathbb{E}_{\theta \sim Q} \log(\frac{Q(\theta)}{P(\theta)}). \tag{4}$$

### 3.3 SGD OPTIMIZATION

Stochastic Gradient Descent (SGD) is a widely adopted method for minimizing the empirical risk $\hat{\mathcal{R}}$. Given a training dataset of size $N$, a mini-batch $\mathcal{S}$ consists of a subset of $S$ sampled independently and identically (i.i.d.) from the set of indices $\{1, \ldots, N\}$. The update rule for SGD can be formally expressed as:

$$\begin{aligned} \theta(t + 1) &= \theta(t) - \eta \nabla_{\theta(t)} \hat{\mathcal{R}}(\theta(t)) \\ &= \theta(t) - \eta \frac{1}{S} \sum_{s \in \mathcal{S}} \nabla_{\theta(t)} F_s(\theta(t)), \end{aligned} \tag{5}$$

where $\eta$ denotes the learning rate and $\nabla_{\theta(t)} \hat{\mathcal{R}}(\theta(t))$ represents the estimated gradient of the empirical risk calculated over the mini-batch $\mathcal{S}$.

## 4 THEORETICAL ANALYSIS OF THE PERFORMANCE GAP BETWEEN FEDERATED AND CENTRALIZED LEARNING

In this section, we develop theoretical foundations for the performance gap between federated and centralized settings and identify theoretically feasible approaches to close this gap. The main ingredient of our theory is the expression of this gap in the view of the PAC-Bayesian framework. We derive non-vacuous bounds for this theoretical expression, showing that the performance gap necessarily exists under equal training resources and how this gap varies with the parameters. Further analysis suggests that only the strategy of introducing new clients or adding data to existing clients is possible to close this gap fully. Due to space limitations, we provide detailed proof for each theoretical finding in the Appendix A.1.

## 4.1 PROBLEM SETUP

We compare federated training with centralized training under the equivalent training conditions. Specifically, the same dataset and model are used for training, and the total number of training computations is equal. In a federated scenario, there are $n$ clients, and a central server connects $n$ clients. Each client $i \in \{1, \ldots, n\}$ possesses a local dataset $\mathcal{D}_i$, with the average dataset size denoted as $m = \frac{1}{n} \sum_{i=1}^{n} |\mathcal{D}_i|$. Thus, the total amount of data across all clients is $nm$. Assuming the federated training of deep neural networks iterates $T$ communication rounds, we follow the FedAvg algorithm (McMahan et al., 2017) to formulate the training process in round $j \in \{1, \ldots, T\}$ as:

$$\bar{\theta}_i(j) = \frac{1}{n} \sum_{i=1}^{n} \theta_i(j) \tag{6}$$

$$\theta_i(j+1) = \bar{\theta}_i(j) - \eta \nabla_{\bar{\theta}_i(j)} \mathbb{E}_{\zeta_i \sim \mathcal{D}_i} F(\bar{\theta}_i(j); \zeta_i). \tag{7}$$

Eq.(6) describes the model aggregation and update process performed on the central server, while Eq.(7) explains the training of the global model on client $i$ using its local dataset $\mathcal{D}_i$. Since the training is carried out using SGD algorithms, we define the local batch size as $k_{Fed}m$, where $\frac{1}{m} \leq k_{Fed} \leq 1$, with the number of local training epochs set to a positive integer $t$. In contrast, the centralized scenario works with a dataset $\mathcal{D} = \bigcup_{i=1}^{n} \mathcal{D}_i$ of total size $D = nm$, and the initial model weights are identical to those used in the federated scenario, expressed as $\{\theta(0) = \theta_i(0) | i \in n\}$. The process of centralized training follows the update rule of SGD described in Eq.(5) and is run for $\frac{T}{n}$ iterations to ensure that the total training compute matches that of the federated scenario. In each iteration, the model $\theta$ is trained with mini-batches of size $k_{Cen}D$ sampled from $\mathcal{D}$ for $t$ epochs, where $\frac{1}{D} \leq k_{Cen} \leq 1$. Additionally, throughout this paper, we assume the constant learning rate $\eta$ and the same batch size for each training scenario, expressed as $S = k_{Fed}m = k_{Cen}D$.

## 4.2 PAC-BAYESIAN GENERALIZATION GAP

To derive the PAC-Bayesian view of the performance gap between federated learning and centralized learning, we first need to establish the PAC-Bayes upper bounds for the generalization error of models trained in each scenario. Similar to the previous studies (Stephan et al., 2017; He et al., 2019), we make some assumptions on SGD to help our proof.

**Assumption 1.** *Assuming all the gradients $\{\nabla_\theta F_s(\theta)\}$ computed from individual training samples are uniformly drawn from a Gaussian distribution whose center is the gradient of the expected risk $g(\theta)$ and the covariance matrix is $C$, expressed as below:*

$$\nabla_\theta F_s(\theta) \sim \mathcal{N}(g(\theta), C), \tag{8}$$

*the stochastic gradients $\hat{g}_s(\theta) = \nabla_{\theta(t)} \hat{\mathcal{R}}(\theta(t))$ calculated from the mini-batches will be assumed to be uniformly sampled from the following Gaussian distribution:*

$$\hat{g}_s(\theta) = \frac{1}{S} \sum_{s \in \mathcal{S}} \nabla_\theta F_s(\theta) \sim \mathcal{N}(g(\theta), \frac{1}{S}C). \tag{9}$$

*Here, this constant matrix $C$ can be further factorized as $C = BB^\intercal$ as covariance matrices are (semi) positive-definite.*

We justify Assumption 1 by the central limit theorem when the training data size is substantially larger than the batch size. Since deep neural networks are typically trained on large-scale datasets in real-world applications, this assumption is generally valid (Weinan, 2017; Stephan et al., 2017).

**Assumption 2.** *Assuming the loss function $F(\theta)$ is smooth, the stationary distribution of the iterates is confined to a local region near a minimum, where the loss is well approximated by a quadratic function with the following form:*

$$F(\theta) = \frac{1}{2} \theta^\intercal A \theta. \tag{10}$$

*where $A$ is the Hessian matrix around the minimum and is (semi) positive-definite.*

Assumption 2 makes sense when SGD converges to a low-variance quasi-stationary distribution near a deep local minimum, where the gradient noise is small compared to the average gradient. Thus

SGD follows a relatively directed path toward the optimum. This assumption is also supported by empirical evidence (see p.1, Figures 1(a) and 1(b) and p.6, Figures 4(a) and 4(b) in (Li et al., 2018)). Additionally, without loss of generality, we assume the global minimum of the loss function is 0 when $\theta = 0$. General cases can be obtained through translation operations, which would not modify the geometry of objective function and its associated generalization ability.

Under Assumption 1, the SGD iterations can be re-expressed in the form of the Ornstein-Uhlenbeck process (Uhlenbeck & Ornstein, 1930):

$$\theta(t+1) - \theta(t) = -\eta \hat{g}_s(\theta(t)) = -\eta g(\theta) + \frac{\eta}{S} B \Delta W, \Delta W \sim \mathcal{N}(0, I). \tag{11}$$

For Eq.(11), the results of the Ornstein-Uhlenbeck process suggest that there exists an analytic stationary distribution in terms of the normalizer $M$ and the matrix $\Sigma$, defined as below:

$$q(\theta) = M \exp \left\{ -\frac{1}{2} \theta^\intercal \Sigma \theta \right\}. \tag{12}$$

Then, based on the above equations and assumptions, we derive a generalization bound for the models trained by federated SGD optimization.

**Theorem 1.** *For any positive real number $\delta \in (0, 1)$, with probability at least $1 - \delta$ over a decentralized training dataset of total size $nm$ across $n$ clients, the following inequality holds for the distribution $Q_{Fed}$ of the output hypothesis learned by federated SGD:*

$$R(Q_{Fed}) - \hat{R}(Q_{Fed})$$

$$\leq \sqrt{\frac{-\log(\det(\Sigma_{Fed})) + \frac{T\eta}{2k_{Fed}m} tr(\bar{C}\bar{A}^{-1}) - d + 2\log(\frac{1}{\delta}) + 2\log(nm) + 4}{4nm - 2}}. \tag{13}$$

*where $C_i$ is the covariance of the loss gradients and $A_i$ is Hessian matrix around the minimum of the loss function for local training on client $i$, $\bar{C} = \frac{1}{n}\sum_{i=1}^{n} C_i$, $\bar{A} = \frac{1}{n}\sum_{i=1}^{n} A_i$, $d$ is the dimension of the model parameter $\theta$ (parameter size), $T$ is the number of communication rounds, $\eta$ is the learning rate and $tr(\bar{C}\bar{A}^{-1})$ is the trace of the product matrix $\bar{C}\bar{A}^{-1}$.*

**Proof Sketch.** The proof of Theorem 1 has three parts. At the beginning, we utilize the update rule of federated training (Eqs.(6) and (7)) and the results of the Ornstein-Uhlenbeck process (Eq.(11)) to find the following stationary solution for the iterates of federated SGD optimization:

$$\theta_{Fed}(T) = \frac{1}{n}\sum_{i=1}^{n} \theta_i(T) = \theta_i(0)e^{-T\bar{A}t} + T\sqrt{\frac{\eta}{k_{Fed}m}} \int_0^t e^{-T\bar{A}(t-t')} \bar{B} dW(t'). \tag{14}$$

Next, according to Eqs.(12) and (14), the property $T\bar{A}\Sigma_{Fed} + \Sigma_{Fed}T\bar{A} = \frac{T^2\eta}{k_{Fed}m}\bar{C}$ is proved. Finally, by assuming that the prior distribution $P$ is a Gaussian or Uniform distribution and combining this property with Lemma 1, we derive a PAC-Bayes upper bound for the generalization error of models trained in federated settings. Note that this bound does not include the number of local training epochs $t$ as $t$ is simplified through integral operations in the proofs (see appendix for details).

By a similar approach, the generalization bound for centralized training under equal training resources can also be proved as follows.

**Corollary 1.** *For any positive real number $\delta \in (0, 1)$, with probability at least $1 - \delta$ over a centralized training dataset of total size $D$ on server, the following inequality holds for the distribution $Q_{Cen}$ of the output hypothesis learned by centralized SGD:*

$$R(Q_{Cen}) - \hat{R}(Q_{Cen})$$

$$\leq \sqrt{\frac{-\log(\det(\Sigma_{Cen})) + \frac{T\eta}{2nk_{Cen}D} tr(CA^{-1}) - d + 2\log(\frac{1}{\delta}) + 2\log(D) + 4}{4D - 2}}. \tag{15}$$

*where $C$ and $A$ are the covariance and Hessian matrix for training with the centralized dataset, and $\Sigma_{Cen}$ is the covariance matrix for the stationary distribution of this global training.*

Since the covariance matrix $C$, the Hessian matrix $A$, and the constant matrix $\Sigma$ are from the stationary distribution of the SGD optimization, it is easy to see that the comparison of two bounds becomes intractable without the knowledge of how these matrices vary by changes in training setup. Therefore, we further present two assumptions and study a special case of the generalization bound.

**Assumption 3.** *We assume that $A$ and $\Sigma$ are symmetric matrices satisfying $A\Sigma = \Sigma A$.*

Assumption 3 implies that the local geometry around the global minimum and the stationary distribution are homogeneous across all dimensions of the parameter space. A similar assumption has also been used in previous papers (He et al., 2019; Jastrzkebski et al., 2017).

**Assumption 4.** *Under the fair comparison condition that the same training dataset is used for both training scenarios, the average data distribution $\bar{\mathcal{D}}$ across $n$ clients of size $m$ is assumed to be independently and identically (i.i.d.) drawn from the global dataset $\mathcal{D}$ of size $D = nm$ in centralized settings and the following properties are satisfied:*

$$\bar{A} \approx A, \quad \bar{C} \approx \frac{1}{n^\gamma}C \tag{16}$$

*where $\gamma$ is a constant that $\gamma > 1$.*

Assumption 4 could be justified by the central limit theorem when the average data size $m$ across clients and the size of global dataset $D$ are both large enough. With the two new assumptions, we can quantify the distance between the above generalization bounds and derive the below theorem.

**Theorem 2.** *When all the above assumptions hold and the training resources for federated and centralized learning are equal, the generalization gap between the models trained through federated SGD optimization and the models trained through centralized SGD optimization has the following analytic solution:*

$$\mathcal{G}_{Fed} - \mathcal{G}_{Cen} = \frac{d\log(\frac{2n^\gamma k_{Fed}m}{T\eta}) + \frac{T\eta}{2n^\gamma k_{Fed}m}tr(CA^{-1}) - d\log(\frac{2nk_{Cen}D}{T\eta}) - \frac{T\eta}{2nk_{Cen}D}tr(CA^{-1})}{4D - 2}. \tag{17}$$

*where $\mathcal{G}$ is the generalization bound of a learning algorithm.*

**Proof Sketch.** The first part of this proof is to re-formulate the generalization bound derived for each training scenario. Based on Assumption 3, we re-arrange the properties found in the proofs of Theorem 1 and Corollary 1 to find an analytic solution for the constant matrix $\Sigma$. Substituting this solution to Eqs.(13) and (15) and applying Assumption 4 will yield new generalization bounds. We then complete the proof by computing the distance between the two new PAC-Bayes upper bounds and re-arranging this distance equation.

Theorem 2 shows the analytic solution of the performance gap in the PAC-Bayesian framework.

### 4.3 THE NON-VACUOUS BOUNDS ON PERFORMANCE GAP

In this subsection, we continue to explore this theoretical expression to gain a deeper understanding of the performance gap. As pointed out at the beginning of the paper, our interest lies in these questions: 1) does the performance gap always exist with equal training resources? 2) how is this gap affected by the environmental variables in the federated scenario? We answer these questions using the following theorem.

**Theorem 3.** *When all conditions of Theorem 2 hold, and assuming that the training resources are equal for both federated and centralized scenarios, the generalization gap between models trained using federated SGD and those trained using centralized SGD satisfies the following inequalities:*

$$\frac{d\log(3^{\gamma-1}) + \frac{(1-3^{\gamma-1})T\eta}{2*3^\gamma k_{Cen}D}tr(CA^{-1})}{4D - 2} \leq \mathcal{G}_{Fed} - \mathcal{G}_{Cen} \leq \frac{d\log(D^{\gamma-1}) + \frac{(1-D^{\gamma-1})T\eta}{2k_{Cen}D^{\gamma+1}}tr(CA^{-1})}{4D - 2}, \tag{18}$$

*for $3 \leq n \leq D$, where $n$ represents the number of clients and $D$ is the total data size across clients. Additionally, when $n = 2$, for any constant $\gamma \gtrsim 1.284$, the generalization gap between federated and centralized training satisfies the following inequality:*

$$\mathcal{G}_{Fed} - \mathcal{G}_{Cen} \geq \frac{d\log(2^{\gamma-1}) + \frac{(1-2^{\gamma-1})T\eta}{2^{\gamma+1}k_{Cen}D}tr(CA^{-1})}{4D - 2}. \tag{19}$$

**Proof Sketch.** We start by proving that the performance gap monotonically increases with $n$ if the condition $n \geq {}^{\gamma-1}\!\!\sqrt{\gamma}$ holds and ${}^{\gamma-1}\!\!\sqrt{\gamma}$ is upper bounded by $e$. Therefore, this monotonic impact

will always hold for $n \geq 3$. By substituting this range of $n$ into Eq.(17), we derive the bound of the performance gap for $n \geq 3$. Next, considering that the parameter $n$ satisfies $\{2 \leq n \leq D | n \in \mathbb{Z}\}$, we compare the performance gap under $n = 2$ with the gap under $n = 3$ to figure out the exact lower bound. The results show that the lower bound for $n = 2$ can only be found with $\gamma \gtrapprox 1.284$.

Theorem 3 establishes non-vacuous bounds for the performance gap between two training scenarios. In Eq.(18), both lower and upper bounds contain two terms in the numerator. The left term can be regarded as a static one capturing the entropy of the gap, and the right term can be considered an empirical one affected throughout the training process. The static term indicates that the gap already exists when two training scenarios are provided with equal training resources, no matter the best and worst case. Moreover, this default gap increases with the model size $d$ and the number of clients $n$. On the other side, the empirical term shows that the gap is decreased through training, but the reduced distance seems quite limited. In the worst case, the denominator of the empirical term contains $D^{\gamma+1}$. Since $D$ represents the total data size, we know that the value of the empirical term in the worst case is extremely small. Similarly, increasing the total data size $D$ cannot completely close the performance gap because the lower bound contains $D$ in the denominator of the empirical term, and the upper bound contains $D$ in both the static and empirical terms.

## 4.4 Strategies for Bridging the Gap

The above theoretical results demonstrate that the performance gap cannot be eliminated completely as long as equal training resources are provided for two scenarios. Therefore, if we still look forward to federated training catching up with centralized training, the federated scenario has to be allowed with an advantage in some training resources. Generally, increasing the data size and model size can result in an improvement in model performance. For example, researchers have concluded scaling laws indicating that the performance of large language models is related to these two parameters (Kaplan et al., 2020; Hoffmann et al., 2022). Besides, previous federated studies have also empirically shown that increasing the number of communication rounds or the number of clients also leads to improved model performance (McMahan et al., 2017; Zhuang et al., 2021). So, we study the related parameters $n$, $m$, $d$, and $T$ in federated settings with a reasonable assumption to identify which one has the potential to close the gap.

**Assumption 5.** *In federated scenarios, the parameter size $d$ of deep neural networks are large enough to satisfy $d > \frac{\log(\det(CA^{-1})\delta^2)}{\log(\frac{2nk_{Fed}m}{T\eta})-1}$ for any real number $\delta \in (0, 1)$, and the number of clients $n$ are also large enough to satisfy $n \geq \sqrt[\gamma-1]{e}$ for any constant $\gamma > 1$, where $m$ is the average data size, $C$ is the magnitude of loss gradient noise and $A$ is the Hessian matrix.*

Assumption 5 basically holds, as deep neural networks are typically over-parameterized to achieve impressive performance (Kaplan et al., 2020; Hoffmann et al., 2022) and realistic federated scenarios often involve a significant amount of clients (Kairouz et al., 2021). When this assumption is valid, the lower and upper bounds in Theorem 3 are both positive, indicating that federated training is inferior to centralized training in generalization. Then, we propose the following theorem.

**Theorem 4.** *When all the above assumptions hold and assuming that the federated scenario is provided with an advantage in training conditions, the following inequalities hold for the generalization gap between models trained through federated SGD and those trained through centralized SGD:*

$$\lim_{n \to \infty} \tilde{\mathcal{G}}_{Fed} - \mathcal{G}_{Cen} \leq 0; \quad \lim_{m \to \infty} \tilde{\mathcal{G}}_{Fed} - \mathcal{G}_{Cen} \leq 0;$$
$$\lim_{d \to \infty} \tilde{\mathcal{G}}_{Fed} - \mathcal{G}_{Cen} = \infty; \quad \lim_{T \to \infty} \tilde{\mathcal{G}}_{Fed} - \mathcal{G}_{Cen} = \infty. \tag{20}$$

*where $\tilde{\mathcal{G}}_{Fed}$ is the generalization bound for federated scenarios having an advantage in training.*

**Proof Sketch.** The proof of Theorem 4 consists of four parts. In each part, we select a parameter and re-establish the theoretical representation of the performance gap by considering that the federated scenario has an advantage in this parameter. Then, we derive a bound for this new expression and compute the limits of this bound when the selected parameter approaches infinity.

Theorem 4 shows us that this performance gap is only likely to be fully closed by 1) introducing new clients or 2) adding data to existing clients. Furthermore, we can also understand from Eq.(20) that the complete close of the performance gap is not feasible by increasing the model size or the number of communication rounds without adding new data.

## 5 EMPIRICAL VALIDATION

### 5.1 EXPERIMENT SETUP

To empirically validate our theoretical findings and ensure that they can be applied to any case, we conduct extensive experiments on different models and datasets. The model architectures we have used are ResNet-18 (He et al., 2016) and Vision Transformer (ViT) (Dosovitskiy et al., 2020), which represent two dominant types of deep neural networks: Convolutional Neural Networks (CNNs) (LeCun et al., 1998), and Transformers (Vaswani et al., 2017). We build 10 models of different sizes for each architecture to study the impact of the model size. On the other hand, we exploit two standard datasets for evaluating the training in different setups: CIFAR-10 (Krizhevsky et al., 2009) with 50000 training images and 10000 validation images in 10 classes, and Mini-ImageNet (Vinyals et al., 2016) with 60000 images in 100 classes extracted from ImageNet (Deng et al., 2009). Since the Mini-ImageNet dataset does not provide a training set with all classes of images, we randomly split it into 48000 training images and 12000 validation images. The complete training set of these datasets will be used in centralized training. To simulate federated scenarios with $n$ clients, we follow our problem setup to divide each training set into $n$ partitions by i.i.d distribution, so each client contains an equal amount of training data for all categories. Furthermore, the batch size and learning rate are kept the same for both setups based on our problem setup. Our codes for experiments were implemented using the PyTorch framework and executed on a server with 8 NVIDIA® RTX A5000 GPUs. The detailed experiment settings and server configuration are provided in the Appendix A.2 due to page limitations.

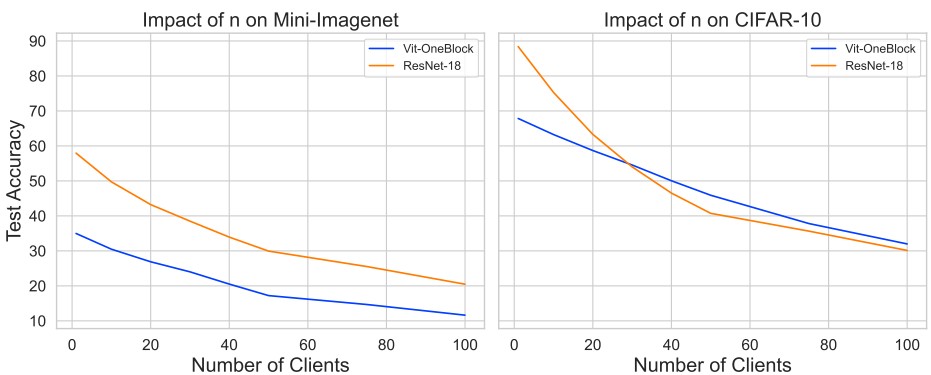

Figure 1: Impact of the number of clients $n$ on the generalization performance. Different colors represent different model architectures. **(Left)** Curves of Mini-ImageNet testing accuracy (%) to the number of clients. **(Right)** Curve of CIFAR-10 testing accuracy (%) to the number of clients. For the centralized scenario, we consider that it corresponds to the case $n = 1$.

### 5.2 EMPIRICAL EVIDENCE

#### 5.2.1 PERFORMANCE GAP UNDER EQUAL TRAINING RESOURCE

We verify our non-vacuous bounds about the performance gap by constructing federated and centralized scenarios with equivalent training resources based on our problem setup. In Eq.(19), the static term contains the number of clients $n$ and the model size $d$. Figure 1 shows that the testing accuracy of models decreases with the number of clients. Since the centralized scenario can be considered as containing only one client (which is the server), the impact of $n$ on the performance gap is justified. On the other hand, we can observe from Figure 2 that the performance gap under equal training resources also increases with the parameter size, which validates our theoretical insights about $d$.

#### 5.2.2 BRIDGE PERFORMANCE GAP BY INCREASING TRAINING RESOURCES

To empirically investigate our theoretical insights about the complete elimination of the performance gap, we designed four sets of experiments for the four parameters involved in Theorem 4. In each

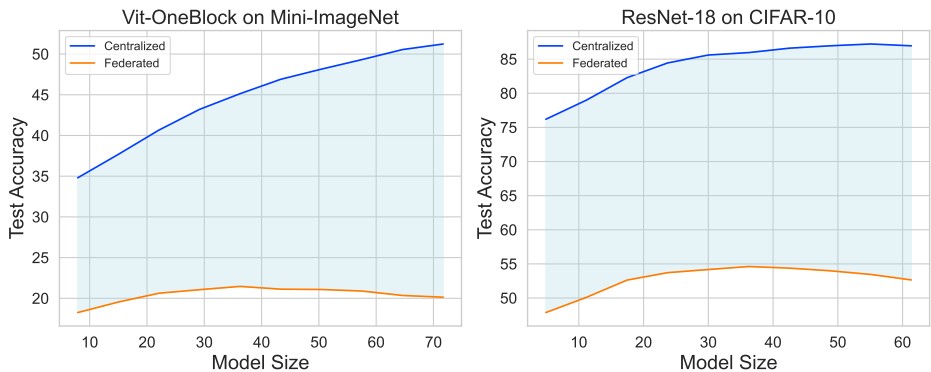

Figure 2: Impact of the model size $d$ on the generalization performance. The performance gap between federated and centralized training is demonstrated by the light-blue area between two lines.

experiment, a centralized scenario is compared with a federated scenario that holds an advantage in one kind of training resource. We gradually amplify this advantage to check if the performance gap can be progressively closed. Due to page limitations, we can only show the experiment results evaluating the strategy of incorporating new clients or adding data to existing clients. Other experimental results giving the federated scenario an advantage over $d$ and $T$ can be found in the Appendix A.3. The results presented in Figure 3 validate Theorem 4. Specifically, we can discover that the generalization performance of models trained in federated setups catches up or surpasses those trained in centralized setups by applying these two strategies.

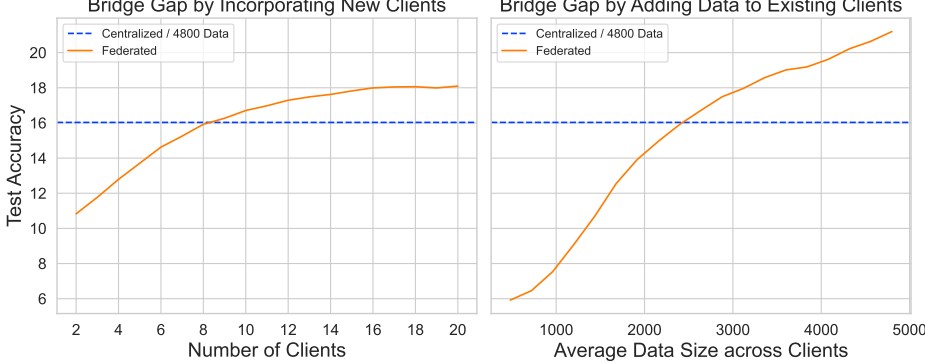

Figure 3: Empirical evidence for fully closing the performance gap between federated and centralized training setup. **(Left)** The strategy of incorporating new clients (increasing the number of clients $n$). **(Right)** The strategy of adding data to existing clients (increasing the average data amount $m$).

## 6 CONCLUSION

This paper re-studies the problem that models trained in federated setups do not perform as well as models trained in centralized setups, focusing on the theoretical exploration of this generalization gap and valid strategies to bridge it. By formulating the gap as the distance between the PAC-Bayes generalization bounds of two scenarios, we derive non-vacuous bounds on this gap and find that it is affected by the training settings and necessarily exists when both scenarios are allocated with equivalent training resources. Therefore, we further consider the case that the federated scenario holds an advantage in training resources and prove that the gap can be closed by introducing new clients or adding data to existing clients, while strategies like increasing model size or communication rounds are not feasible. In addition, extensive experiments are conducted to empirically analyze the performance gap. The experimental results are fully aligned with our theoretical findings.

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

## A APPENDIX

### A.1 FULL PROOFS FOR THEORETICAL ANALYSIS

At the beginning of the proof, we introduce some necessary lemmas.

**Lemma 2.** *Under the above assumptions, if learning rate $\eta$ and batch size $S = k_{Fed}m$ are fixed, we can derive the following analytic solution for the output parameter $\theta_{Fed}(T)$ of federated SGD:*

$$\theta_{Fed}(T) = \frac{1}{n} \sum_{i=1}^{n} \theta_i(T) = \theta_i(0)e^{-T\bar{A}t} + T\sqrt{\frac{\eta}{k_{Fed}m}} \int_0^t e^{-T\bar{A}(t-t')}\bar{B}dW(t'). \qquad (21)$$

*where $A_i$ is the Hessian matrix and $B_i$ is the covariance matrix for local training on client $i$, respectively. Besides, we have $\bar{A} = \frac{1}{n}\sum_{i=1}^{n} A_i$ and $\bar{B} = \frac{1}{n}\sum_{i=1}^{n} B_i$.*

*Proof.* From the result of the Ornstein-Uhlenbeck process (Uhlenbeck & Ornstein, 1930), the analytical solution for the local SGD training on client $i$ in the first round $j = 1$ is expressed as follows:

$$\theta_i(1) = \theta_i(0)e^{-A_i t} + \sqrt{\frac{\eta}{k_{Fed}m}} \int_0^t e^{-A_i(t-t')} B_i dW(t'), \qquad (22)$$

where $W(t')$ is a white noise and follows $\mathcal{N}(0, I)$. Then based on the update rule of FedAvg defined in Eqs.(6) and (7), the analytic solution for local training on client $i$ in the round $j = 2$ should be:

$$\theta_i(2) = \frac{1}{n} \sum_{i=1}^{n} \theta_i(1)e^{-A_i t} + \sqrt{\frac{\eta}{k_{Fed}m}} \int_0^t e^{-A_i(t-t')} B_i dW(t'). \qquad (23)$$

Substituting Eq.(22) into Eq.(23), we have

$$\theta_i(2) = \frac{1}{n} \sum_{i=1}^{n} \left( \theta_i(0)e^{-A_i t} + \sqrt{\frac{\eta}{k_{Fed}m}} \int_0^t e^{-A_i(t-t')} B_i dW(t') \right) e^{-A_i t}$$

$$+ \sqrt{\frac{\eta}{k_{Fed}m}} \int_0^t e^{-A_i(t-t')} B_i dW(t')$$

$$= \theta_i(0)e^{-2\bar{A}t} + \sqrt{\frac{\eta}{k_{Fed}m}} \int_{-t}^0 e^{-\bar{A}(t-t')} \bar{B} dW(t') + \sqrt{\frac{\eta}{k_{Fed}m}} \int_0^t e^{-A_i(t-t')} B_i dW(t'). \tag{24}$$

In the same way, we formulate the analytic solution in the round $j = 3$ as follows:

$$\theta_i(3) = \frac{1}{n} \sum_{i=1}^{n} (\theta_i(0)e^{-2\bar{A}t} + \sqrt{\frac{\eta}{k_{Fed}m}} \int_{-t}^0 e^{-\bar{A}(t-t')} \bar{B} dW(t')$$

$$+ \sqrt{\frac{\eta}{k_{Fed}m}} \int_0^t e^{-A_i(t-t')} B_i dW(t'))e^{-A_i t} + \sqrt{\frac{\eta}{k_{Fed}m}} \int_0^t e^{-A_i(t-t')} B_i dW(t')$$

$$= \theta_i(0)e^{-2\bar{A}t} \frac{1}{n} \sum_{i=1}^{n} e^{-A_i t} + \sqrt{\frac{\eta}{k_{Fed}m}} \int_{-t}^0 e^{-\bar{A}(t-t')} \bar{B} dW(t') \frac{1}{n} \sum_{i=1}^{n} e^{-A_i t}$$

$$+ \sqrt{\frac{\eta}{k_{Fed}m}} \frac{1}{n} \sum_{i=1}^{n} \int_0^t e^{-A_i(t-t')} e^{-A_i t} B_i dW(t') + \sqrt{\frac{\eta}{k_{Fed}m}} \int_0^t e^{-A_i(t-t')} B_i dW(t')$$

$$= \theta_i(0)e^{-3\bar{A}t} + \sqrt{\frac{\eta}{k_{Fed}m}} \left( \int_{-2t}^{-t} e^{-\bar{A}(t-t')} \bar{B} dW(t') + \int_{-t}^0 e^{-\bar{A}(t-t')} \bar{B} dW(t') \right)$$

$$+ \sqrt{\frac{\eta}{k_{Fed}m}} \int_0^t e^{-A_i(t-t')} B_i dW(t')$$

$$= \theta_i(0)e^{-3\bar{A}t} + \sqrt{\frac{\eta}{k_{Fed}m}} \int_{-2t}^0 e^{-\bar{A}(t-t')} \bar{B} dW(t') + \sqrt{\frac{\eta}{k_{Fed}m}} \int_0^t e^{-A_i(t-t')} B_i dW(t'). \tag{25}$$

Similarly, the analytic solution after $T$ rounds of federated training can be derived as the following equation:

$$\theta_{Fed}(T) = \frac{1}{n}\sum_{i=1}^{n}\theta_i(T)$$

$$= \theta_i(0)e^{-T\bar{A}t} + \sqrt{\frac{\eta}{k_{Fed}m}}\int_{(1-T)t}^{0}e^{-\bar{A}(t-t')}\bar{B}dW(t') + \sqrt{\frac{\eta}{k_{Fed}m}}\frac{1}{n}\sum_{i=1}^{n}\int_{0}^{t}e^{-A_i(t-t')}B_idW(t')$$

$$= \theta_i(0)e^{-T\bar{A}t} + \sqrt{\frac{\eta}{k_{Fed}m}}\int_{(1-T)t}^{0}e^{-\bar{A}(t-t')}\bar{B}dW(t') + \sqrt{\frac{\eta}{k_{Fed}m}}\int_{0}^{t}e^{-\bar{A}(t-t')}\bar{B}dW(t')$$

$$= \theta_i(0)e^{-T\bar{A}t} + \sqrt{\frac{\eta}{k_{Fed}m}}\int_{(1-T)t}^{t}e^{-\bar{A}(t-t')}\bar{B}dW(t')$$

$$= \theta_i(0)e^{-T\bar{A}t} + \sqrt{\frac{\eta}{k_{Fed}m}}\frac{1-e^{-T\bar{A}t}}{\bar{A}}\bar{B}$$

$$= \theta_0 e^{-T\bar{A}t} + \sqrt{\frac{\eta}{k_{Fed}m}}\frac{T(1-e^{-T\bar{A}t})}{T\bar{A}}\bar{B}$$

$$= \theta_0 e^{-T\bar{A}t} + T\sqrt{\frac{\eta}{k_{Fed}m}}\int_{0}^{t}e^{-T\bar{A}(t-t')}\bar{B}dW(t'),$$

$$(26)$$

which completes the proof. $\qquad\square$

**Lemma 3.** *Under the Assumption 2, the stationary distribution of the Ornstein-Uhlenbeck process for the federated SGD,*

$$q(\theta_{Fed}) = M\exp\left\{-\frac{1}{2}\theta_{Fed}^{\mathsf{T}}\Sigma_{Fed}^{-1}\theta\right\},\tag{27}$$

*has the following property,*

$$T\bar{A}\Sigma_{Fed} + \Sigma_{Fed}T\bar{A} = \frac{T^2\eta}{k_{Fed}m}\bar{C}.\tag{28}$$

*where $M$ is the normalizer and $\Sigma_{Fed}$ is the covariance matrix of the stationary distribution.*

*Proof.* From Eq.(27), we know that

$$\Sigma_{Fed} = \mathbb{E}_{\theta\sim Q}[\theta_{Fed}\theta_{Fed}^{\mathsf{T}}].\tag{29}$$

Then, according to Eq.(26), we can derive the following equation:

$$T\bar{A}\Sigma_{Fed} + \Sigma_{Fed}T\bar{A} = \frac{T^2\eta}{k_{Fed}m}\int_{-\infty}^{t}T\bar{A}e^{-T\bar{A}(t-t')}\bar{C}e^{-T\bar{A}(t-t')}dt'$$

$$+ \frac{T^2\eta}{k_{Fed}m}\int_{-\infty}^{t}e^{-T\bar{A}(t-t')}\bar{C}e^{-T\bar{A}(t-t')}dt'T\bar{A}$$

$$= \frac{T^2\eta}{k_{Fed}m}\int_{-\infty}^{t}\frac{d}{dt'}(e^{-T\bar{A}(t-t')}\bar{C}e^{-T\bar{A}(t-t')})$$

$$= \frac{T^2\eta}{k_{Fed}m}\bar{C},$$

$$(30)$$

which completes the proof. $\qquad\square$

### A.1.1 PROOF OF THEOREM 1

*Proof.* Following the classical Pac-Bayesian framework, we suppose the prior distribution over the parameter space $\theta$ is $P$, and the distribution of the learned hypothesis from the federated SGD algorithm is $Q$. Then according to Eq.(27), the densities of the stationary distribution $Q$ and the

prior distribution $P$ are respectively $q(\theta)$ and $p(\theta)$ in terms of the parameter $\theta$ and can be expressed as the following equations:

$$q(\theta) = \frac{1}{\sqrt{2\pi \det(\Sigma_{Fed})}} \exp\left\{-\frac{1}{2}\theta^\mathsf{T}\Sigma_{Fed}^{-1}\theta\right\},$$

$$p(\theta) = \frac{1}{\sqrt{2\pi \det(I)}} \exp\left\{-\frac{1}{2}\theta^\mathsf{T}I\theta\right\}. \tag{31}$$

Thus we have

$$\log\left(\frac{q(\theta)}{p(\theta)}\right) = \log\left(\frac{\sqrt{2\pi \det(I)}}{\sqrt{2\pi \det(\Sigma_{Fed})}} \exp\left\{\frac{1}{2}\theta^\mathsf{T}I\theta - \frac{1}{2}\theta^\mathsf{T}\Sigma_{Fed}^{-1}\theta\right\}\right)$$

$$= \frac{1}{2}\log\left(\frac{1}{\det(\Sigma_{Fed})}\right) + \frac{1}{2}\left(\theta^\mathsf{T}I\theta - \theta^\mathsf{T}\Sigma_{Fed}^{-1}\theta\right). \tag{32}$$

Here, we can calculate the KL divergence between the distribution $Q$ and $P$ by applying Eq.(4) in Lemma 1:

$$D(Q\|P) = \mathbb{E}_{\theta\sim Q}\left(\log\frac{Q(\theta)}{P(\theta)}\right)$$

$$= \int_{\theta\in\Theta} \log\left(\frac{q(\theta)}{p(\theta)}\right) q(\theta)d\theta$$

$$= \int_{\theta\in\Theta} \left[\frac{1}{2}\log\left(\frac{1}{\det(\Sigma_{Fed})}\right) + \frac{1}{2}\left(\theta^\mathsf{T}I\theta - \theta^\mathsf{T}\Sigma_{Fed}^{-1}\theta\right)\right] q(\theta)d\theta$$

$$= \frac{1}{2}\log\left(\frac{1}{\sqrt{\det(\Sigma_{Fed})}}\right) + \frac{1}{2}\int_{\theta\in\Theta} \theta^\mathsf{T}I\theta q(\theta)d\theta - \frac{1}{2}\int_{\mathbb{R}^{|\mathcal{S}|}} \theta^\mathsf{T}\Sigma_{Fed}^{-1}q(\theta)d\theta \tag{33}$$

$$= \frac{1}{2}\log\left(\frac{1}{\sqrt{\det(\Sigma_{Fed})}}\right) + \frac{1}{2}\mathbb{E}_{\theta\sim\mathcal{N}(0,\Sigma_{Fed})}\theta^\mathsf{T}I\theta - \frac{1}{2}\mathbb{E}_{\theta\sim\mathcal{N}(0,\Sigma_{Fed})}\theta^\mathsf{T}\Sigma_{Fed}^{-1}\theta$$

$$= \frac{1}{2}\log\left(\frac{1}{\sqrt{\det(\Sigma_{Fed})}}\right) + \frac{1}{2}\mathrm{tr}(\Sigma_{Fed} - I).$$

Since we have proved from Lemma 3 that $T\bar{A}\Sigma_{Fed} + \Sigma_{Fed}T\bar{A} = \frac{T^2\eta}{k_{Fed}m}\bar{C}$, we have

$$\bar{A}\Sigma_{Fed}\bar{A}^{-1} + \Sigma_{Fed} = \frac{T^2\eta}{Tk_{Fed}m}\bar{C}\bar{A}^{-1}$$

$$\mathrm{tr}(\bar{A}\Sigma_{Fed}\bar{A}^{-1} + \Sigma_{Fed}) = \mathrm{tr}(\frac{T\eta}{k_{Fed}m}\bar{C}\bar{A}^{-1}). \tag{34}$$

For the left hand side, we can change it to the following equation:

$$\mathrm{LHS} = \mathrm{tr}(\bar{A}\Sigma_{Fed}\bar{A}^{-1} + \Sigma_{Fed})$$

$$= \mathrm{tr}(\bar{A}\Sigma_{Fed}\bar{A}^{-1}) + \mathrm{tr}(\Sigma_{Fed})$$

$$= \mathrm{tr}(\bar{A}\bar{A}^{-1}\Sigma_{Fed}) + \mathrm{tr}(\Sigma_{Fed}) \tag{35}$$

$$= \mathrm{tr}(\Sigma_{Fed}) + \mathrm{tr}(\Sigma_{Fed})$$

$$= 2\mathrm{tr}(\Sigma_{Fed}).$$

Therefore,

$$\mathrm{tr}(\Sigma_{Fed}) = \frac{1}{2}\mathrm{tr}(\frac{T\eta}{k_{Fed}m}\bar{C}\bar{A}^{-1}) = \frac{T\eta}{2k_{Fed}m}\mathrm{tr}(\bar{C}\bar{A}^{-1}). \tag{36}$$

On the other side, we can simply calculate that $\mathrm{tr}(I) = d$, because $I \in \mathbb{R}^{d\times d}$, where $d$ is the dimension of the parameter $\theta$. Then we can have

$$D(Q_{Fed}\|P) = -\frac{1}{2}\log(\det(\Sigma_{Fed})) + \frac{1}{2}\mathrm{tr}(\Sigma_{Fed}) - \frac{1}{2}\mathrm{tr}(I)$$

$$= -\frac{1}{2}\log(\det(\Sigma_{Fed})) + \frac{T\eta}{4k_{Fed}m}\mathrm{tr}(\bar{C}\bar{A}^{-1}) - \frac{1}{2}d. \tag{37}$$

By inserting the Eq.(37) into Eq.(3), we can drive the following inequality for the global training sample set of size $nm$:

$$R(Q_{Fed}) - \hat{R}(Q_{Fed})$$
$$\leq \sqrt{\frac{-\log(\det(\Sigma_{Fed})) + \frac{T\eta}{2k_{Fed}m}\text{tr}(\bar{C}\bar{A}^{-1}) - d + 2\log(\frac{1}{\delta}) + 2\log(nm) + 4}{4nm - 2}}, \tag{38}$$

which has completed the proof. □

**Lemma 4.** *Under all assumptions of Lemma 2, if learning rate $\eta$ and batch size $S = k_{Cen}D$ are fixed, we can derive the following analytic solution for the output parameter of centralized SGD trained on the same amount of training data:*

$$\theta_{Cen}(T) = \theta(0)e^{-\frac{T}{n}At} + \frac{T}{n}\sqrt{\frac{\eta}{k_{Cen}D}}\int_0^t e^{-\frac{T}{n}A(t-t')}BdW(t')). \tag{39}$$

*where $A$ is the Hessian matrix and $B$ is the covariance matrix for training on the centralized dataset of size $D$.*

*Proof.* Based on Eq.(5) and the result of the Ornstein-Uhlenbeck process (Uhlenbeck & Ornstein, 1930), we can simply derive the following analytic solution for the baseline centralized SGD:

$$\theta_{Cen}(T) = \theta(0)e^{-\frac{T}{n}At} + \frac{T}{n}\sqrt{\frac{\eta}{k_{Cen}D}}\int_0^t e^{-\frac{T}{n}A(t-t')}BdW(t')). \tag{40}$$

Thus completing the proof. □

**Lemma 5.** *When Assumption 2 holds, the Ornstein-Uhlenbeck process's stationary distribution for the baseline centralized SGD,*

$$q(\theta_{Cen}) = M\exp\left\{-\frac{1}{2}\theta^{\mathsf{T}}\Sigma_{Cen}^{-1}\theta\right\}, \tag{41}$$

*has the following property,*

$$\frac{T}{n}A\Sigma_{Cen} + \Sigma_{Cen}\frac{T}{n}A = \frac{T^2\eta}{n^2 k_{Cen}D}C. \tag{42}$$

*Proof.* Based on Eq.(41), we know that

$$\Sigma_{Cen} = \mathbb{E}_{\theta\sim Q}[\theta_{Cen}\theta_{Cen}^{\mathsf{T}}]. \tag{43}$$

Then, by combining Eq.(39) and Eq.(43), we can derive the following equation:

$$\begin{aligned}
\frac{T}{n}A\Sigma_{Cen} + \Sigma_{Cen}\frac{T}{n}A &= \frac{T^2\eta}{n^2 k_{Cen}D}\int_{-\infty}^t \frac{T}{n}Ae^{-\frac{T}{n}A(t-t')}Ce^{-\frac{T}{n}A(t-t')}dt' \\
&\quad + \frac{T^2\eta}{n^2 k_{Cen}D}\int_{-\infty}^t e^{-\frac{T}{n}A(t-t')}Ce^{-\frac{T}{n}A(t-t')}dt'\frac{T}{n}A \\
&= \frac{T^2\eta}{n^2 k_{Cen}D}\int_{-\infty}^t \frac{d}{dt'}(e^{-\frac{T}{n}A(t-t')}Ce^{-\frac{T}{n}A(t-t')}) \\
&= \frac{T^2\eta}{n^2 k_{Cen}D}C,
\end{aligned} \tag{44}$$

which completes the proof. □

### A.1.2 PROOF OF COROLLARY 1

*Proof.* Since we have proved from Lemma 5 that $\frac{T}{n}A\Sigma_{Cen} + \Sigma_{Cen}\frac{T}{n}A = \frac{T^2\eta}{n^2 k_{Cen}D}C$, we have

$$
\begin{aligned}
A\Sigma_{Cen} + \Sigma_{Cen}A &= \frac{T\eta}{nk_{Cen}D}C \\
A\Sigma_{Cen}A^{-1} + \Sigma_{Cen} &= \frac{T\eta}{nk_{Cen}D}CA^{-1} \\
\text{tr}(A\Sigma_{Cen}A^{-1} + \Sigma_{Cen}) &= \text{tr}(\frac{T\eta}{nk_{Cen}D}CA^{-1}) \\
2\text{tr}(\Sigma_{Cen}) &= \text{tr}(\frac{T\eta}{nk_{Cen}D}CA^{-1}) \\
\text{tr}(\Sigma_{Cen}) &= \frac{T\eta}{2nk_{Cen}D}\text{tr}(CA^{-1}).
\end{aligned}
\tag{45}
$$

Like the proof of Theorem 1, by substituting the Eq.(45) into Eq.(33), we can compute the KL divergence between the distribution of the output hypothesis and the prior distribution as below:

$$
\begin{aligned}
D(Q_{Cen}||P) &= -\frac{1}{2}\log(\det(\Sigma_{Cen})) + \frac{1}{2}\text{tr}(\Sigma_{Cen}) - \frac{1}{2}\text{tr}(I) \\
&= -\frac{1}{2}\log(\det(\Sigma_{Cen})) + \frac{T\eta}{4nk_{Cen}D}\text{tr}(\bar{C}\bar{A}^{-1}) - \frac{1}{2}d.
\end{aligned}
\tag{46}
$$

According to Lemma 1, then we can derive the following inequality to bound the generalization error of the baseline centralized SGD:

$$
\begin{aligned}
&R(Q_{Cen}) - \hat{R}(Q_{Cen}) \\
&\leq \sqrt{\frac{-\log(\det(\Sigma_{Cen})) + \frac{T\eta}{2nk_{Cen}D}\text{tr}(CA^{-1}) - d + 2\log(\frac{1}{\delta}) + 2\log(D) + 4}{4D - 2}}.
\end{aligned}
\tag{47}
$$

The proof has been completed. $\qquad\square$

### A.1.3 PROOF OF THEOREM 2

*Proof.* Based on Assumption 3 and 4, we can re-formulate Eq.(28) in Lemma 3 to

$$
\begin{aligned}
T\bar{A}\Sigma_{Fed} + \Sigma_{Fed}T\bar{A} &= \frac{T^2\eta}{k_{Fed}m}\bar{C} \\
2T\Sigma_{Fed}\bar{A} &= \frac{T^2\eta}{k_{Fed}m}\bar{C} \\
\Sigma_{Fed} &= \frac{T\eta}{2k_{Fed}m}\bar{C}\bar{A}^{-1} \\
\Sigma_{Fed} &= \frac{T\eta}{2n^\gamma k_{Fed}m}CA^{-1}.
\end{aligned}
\tag{48}
$$

By substituting Eq.(48) into Eq.(38) and applying the Assumption 4, we have

$$
\begin{aligned}
&R(Q_{Fed}) - \hat{R}(Q_{Fed}) \\
&\leq \sqrt{\frac{-\log(\det(\frac{T\eta}{2n^\gamma k_{Fed}m}CA^{-1})) + \frac{T\eta}{2n^\gamma k_{Fed}m}\text{tr}(CA^{-1}) - d + 2\log(\frac{1}{\delta}) + 2\log(nm) + 4}{4nm - 2}} \\
&\leq \sqrt{\frac{-\log((\frac{T\eta}{2n^\gamma k_{Fed}m})^d \det(CA^{-1})) + \frac{T\eta}{2n^\gamma k_{Fed}m}\text{tr}(CA^{-1}) - d + 2\log(\frac{1}{\delta}) + 2\log(nm) + 4}{4nm - 2}} \\
&\leq \sqrt{\frac{d\log(\frac{2n^\gamma k_{Fed}m}{T\eta}) - \log(\det(CA^{-1})) + \frac{T\eta}{2n^\gamma k_{Fed}m}\text{tr}(CA^{-1}) - d + 2\log(\frac{1}{\delta}) + 2\log(nm) + 4}{4nm - 2}} \\
&\leq \frac{d\log(\frac{2n^\gamma k_{Fed}m}{T\eta}) - \log(\det(CA^{-1})) + \frac{T\eta}{2n^\gamma k_{Fed}m}\text{tr}(CA^{-1}) - d + 2\log(\frac{1}{\delta}) + 2\log(nm) + 4}{4nm - 2}
\end{aligned}
\tag{49}
$$

Similarly, according to Assumption 3, we can re-formulate Eq.(42) to:

$$\frac{T}{n}A\Sigma_{Cen} + \Sigma_{Cen}\frac{T}{n}A = \frac{T^2\eta}{n^2 k_{Cen}D}C$$

$$2\Sigma_{Cen}A = \frac{T\eta}{n k_{Cen}D}C \tag{50}$$

$$\Sigma_{Cen} = \frac{T\eta}{2n k_{Cen}D}CA^{-1}.$$

By inserting Eq.(50) into Eq.(47) and re-arranging the equation, we have

$$R(Q_{Cen}) - \hat{R}(Q_{Cen})$$

$$\leq \sqrt{\frac{-\log(\det(\frac{T\eta}{2n k_{Cen}D}CA^{-1})) + \frac{T\eta}{2n k_{Cen}D}\mathrm{tr}(CA^{-1}) - d + 2\log(\frac{1}{\delta}) + 2\log(D) + 4}{4D - 2}}$$

$$\leq \sqrt{\frac{d\log(\frac{2n k_{Cen}D}{T\eta}) - \log(\det(CA^{-1})) + \frac{T\eta}{2n k_{Cen}D}\mathrm{tr}(CA^{-1}) - d + 2\log(\frac{1}{\delta}) + 2\log(D) + 4}{4D - 2}}$$

$$\leq \frac{d\log(\frac{2n k_{Cen}D}{T\eta}) - \log(\det(CA^{-1})) + \frac{T\eta}{2n k_{Cen}D}\mathrm{tr}(CA^{-1}) - d + 2\log(\frac{1}{\delta}) + 2\log(D) + 4}{4D - 2}$$

$$\tag{51}$$

For Eqs.(49) and (51), we define

$$\mathcal{G}_{Fed} = \frac{d\log\left(\frac{2n^\gamma k_{Fed}m}{T\eta}\right) - \log\left(\det\left(CA^{-1}\right)\right) + \frac{T\eta}{2n^\gamma k_{Fed}m}\mathrm{tr}\left(CA^{-1}\right) - d + 2\log\left(\frac{1}{\delta}\right) + 2\log(nm) + 4}{4nm - 2},$$

$$\mathcal{G}_{Cen} = \frac{d\log\left(\frac{2n k_{Cen}D}{T\eta}\right) - \log\left(\det\left(CA^{-1}\right)\right) + \frac{T\eta}{2n k_{Cen}D}\mathrm{tr}\left(CA^{-1}\right) - d + 2\log\left(\frac{1}{\delta}\right) + 2\log(D) + 4}{4D - 2}.$$

$$\tag{52}$$

The difference between $\mathcal{G}_{Fed}$ and $\mathcal{G}_{Cen}$, which is considered as the gap in the generalization performance, can be derived with the following form:

$$\mathcal{G}_{Fed} - \mathcal{G}_{Cen}$$

$$= \frac{d\log(\frac{2n^\gamma k_{Fed}m}{T\eta}) - \log(\det(CA^{-1})) + \frac{T\eta}{2n^\gamma k_{Fed}m}\mathrm{tr}(CA^{-1}) - d + 2\log(\frac{1}{\delta}) + 2\log(nm) + 4}{4nm - 2}$$

$$- \frac{d\log(\frac{2n k_{Cen}D}{T\eta}) - \log(\det(CA^{-1})) + \frac{T\eta}{2n k_{Cen}D}\mathrm{tr}(CA^{-1}) - d + 2\log(\frac{1}{\delta}) + 2\log(D) + 4}{4D - 2}$$

$$= \frac{d\log(\frac{2n^\gamma k_{Fed}m}{T\eta}) + \frac{T\eta}{2n^\gamma k_{Fed}m}\mathrm{tr}(CA^{-1}) - d\log(\frac{2n k_{Cen}D}{T\eta}) - \frac{T\eta}{2n k_{Cen}D}\mathrm{tr}(CA^{-1})}{4D - 2}.$$

$$\tag{53}$$

The proof has been completed.

$\square$

### A.1.4 PROOF OF THEOREM 3

*Proof.* At the beginning, we construct the following helper function:

$$f(n) = d\log\left(\frac{2n^\gamma k_{Cen}D}{T\eta}\right) + \frac{T\eta}{2n^\gamma k_{Cen}D}\mathrm{tr}\left(CA^{-1}\right)$$

$$- d\log\left(\frac{2n k_{Cen}D}{T\eta}\right) - \frac{T\eta}{2n k_{Cen}D}\mathrm{tr}\left(CA^{-1}\right). \tag{54}$$

The derivative of this helper function is:

$$f'(n) = \frac{\gamma d}{n} - \frac{\gamma T\eta}{2n^{\gamma+1}k_{Cen}D}\mathrm{tr}\left(CA^{-1}\right) - \frac{d}{n} + \frac{T\eta}{2n^2 k_{Cen}D}\mathrm{tr}\left(CA^{-1}\right)$$

$$= \frac{(\gamma - 1)d}{n} + \frac{(n^{\gamma-1} - \gamma)T\eta}{2n^{\gamma+1}k_{Cen}D}\mathrm{tr}\left(CA^{-1}\right). \tag{55}$$

For Eq.(55), when $n \geq \sqrt[\gamma-1]{\gamma}$, we have $n^{\gamma-1} - \gamma \geq 0$. Since the constant $\gamma$ satisfies $\gamma > 1$, we can prove $f'(n) > 0$ when $n \geq \sqrt[\gamma-1]{\gamma}$. Then, we construct another helper function and the derivative of this new helper function as follows:

$$g(x) = x^{\frac{1}{x-1}} = e^{\frac{1}{x-1}\log(x)}$$

$$g'(x) = e^{\frac{1}{x-1}\log(x)} \frac{1 - \frac{1}{x} - \log(x)}{(x-1)^2}. \tag{56}$$

From Eq.(56), since $1 - \frac{1}{x} - \log(x) < 0$, it is clear that $g'(x) < 0$. Thus, we have $g(x) < g(1) = e$ and $\sqrt[\gamma-1]{\gamma} < e$. According to Eq.(54), the analytic solution of $\mathcal{G}_{Fed} - \mathcal{G}_{Cen}$ is monotonically increasing with $n$ when $n \geq e$. Because of $n \in \mathbb{Z}^+$, substituting $n = 3$ and $n = D$ into Eq.(53) will derive the following inequalities for $3 \leq n \leq D$:

$$\frac{d\log\left(3^{\gamma-1}\right) + \frac{\left(1-3^{\gamma-1}\right)T\eta}{2*3^\gamma k_{Cen}D}\mathrm{tr}\left(CA^{-1}\right)}{4D-2} \leq \mathcal{G}_{Fed} - \mathcal{G}_{Cen} \leq \frac{d\log\left(D^{\gamma-1}\right) + \frac{\left(1-D^{\gamma-1}\right)T\eta}{2k_{Cen}D^{\gamma+1}}\mathrm{tr}\left(CA^{-1}\right)}{4D-2}. \tag{57}$$

However, the lower bound of $n$ is actually $n = 2$. To find the bound of $\mathcal{G}_{Fed} - \mathcal{G}_{Cen}$ covering the entire range $\{2 \leq n \leq D | n \in Z\}$, we need to compare $f(2)$ with $f(3)$ as follows:

$$f(2) - f(3)$$
$$= -d\log\left(\frac{T\eta}{2^{\gamma+1}k_{Cen}D}\right) + \frac{T\eta}{2^{\gamma+1}k_{Cen}D}\mathrm{tr}\left(CA^{-1}\right) + d\log\left(\frac{T\eta}{4k_{Cen}D}\right) - \frac{T\eta}{4k_{Cen}D}\mathrm{tr}\left(CA^{-1}\right)$$
$$- \left(-d\log\left(\frac{T\eta}{2*3^\gamma k_{Cen}D}\right) + \frac{T\eta}{2*3^\gamma k_{Cen}D}\mathrm{tr}\left(CA^{-1}\right) + d\log\left(\frac{T\eta}{6k_{Cen}D}\right) - \frac{T\eta}{6k_{Cen}D}\mathrm{tr}\left(CA^{-1}\right)\right)$$
$$= d\log\left(2^{\gamma-1}\right) + \frac{\left(1-2^{\gamma-1}\right)T\eta}{2^{\gamma+1}k_{Cen}D}\mathrm{tr}\left(CA^{-1}\right) - d\log\left(3^{\gamma-1}\right) - \frac{\left(1-3^{\gamma-1}\right)T\eta}{2*3^\gamma k_{Cen}D}\mathrm{tr}\left(CA^{-1}\right)$$
$$= (\gamma-1)d\log\left(\frac{2}{3}\right) + \left(\frac{1-2^{\gamma-1}}{2^{\gamma+1}} - \frac{1-3^{\gamma-1}}{3^{\gamma+1}}\right)\frac{T\eta\,\mathrm{tr}\left(CA^{-1}\right)}{k_{Cen}D}. \tag{58}$$

Eq.(58) has two terms. The left term appears to be less than $0$ since $\gamma > 1$. For the right term, we need to solve the condition of $\gamma$ and find that $\frac{1-2^{\gamma-1}}{2^{\gamma+1}} - \frac{1-3^{\gamma-1}}{3^{\gamma+1}} < 0$ when $\gamma \gtrsim 1.284$. By combining the above results, we derive that $f(2) < f(3)$ when $\gamma \gtrsim 1.284$. In summary, when the following condition $\gamma \gtrsim 1.284$ holds, we have

$$\frac{d\log\left(2^{\gamma-1}\right) + \frac{\left(1-2^{\gamma-1}\right)T\eta}{2^{\gamma+1}k_{Cen}D}\mathrm{tr}\left(CA^{-1}\right)}{4D-2} \leq \mathcal{G}_{Fed} - \mathcal{G}_{Cen} \leq \frac{d\log\left(D^{\gamma-1}\right) + \frac{\left(1-D^{\gamma-1}\right)T\eta}{2k_{Cen}D^{\gamma+1}}\mathrm{tr}\left(CA^{-1}\right)}{4D-2} \tag{59}$$

for $\{2 \leq n \leq D | n \in \mathbb{Z}\}$ by substituting the lower bound $n = 2$ and the upper bound $n = D$ into Eq.(54) and re-arranging the results. The proof has been completed. $\qquad\square$

### A.1.5 PROOF OF THEOREM 4

*Proof.* We define $\tilde{\mathcal{G}}_{Fed}$ for the generalization bound of federated scenarios having an advantage in training resources and start with the case of $n$ tends to infinity. The performance gap $\tilde{\mathcal{G}}_{Fed} - \mathcal{G}_{Cen}$ for this case is formulated as the below form:

$$\tilde{\mathcal{G}}_{Fed} - \mathcal{G}_{Cen}$$
$$= \frac{d\log\left(\frac{2n^\gamma k_{Fed}m}{T\eta}\right) - \log\left(\det\left(CA^{-1}\right)\right) + \frac{T\eta}{2n^\gamma k_{Fed}m}\mathrm{tr}\left(CA^{-1}\right) - d + 2\log\left(\frac{1}{\delta}\right) + 2\log\left(nm\right) + 4}{4nm-2}$$
$$- \frac{d\log\left(\frac{2nk_{Cen}D}{T\eta}\right) - \log\left(\det\left(CA^{-1}\right)\right) + \frac{T\eta}{2nk_{Cen}D}\mathrm{tr}\left(CA^{-1}\right) - d + 2\log\left(\frac{1}{\delta}\right) + 2\log\left(D\right) + 4}{4D-2}. \tag{60}$$

According to Assumption 5, we have

$$
d > \frac{\log(\det(CA^{-1})\delta^2)}{\log(\frac{2nk_{Fed}m}{T\eta}) - 1}
$$

$$
d(\log(\frac{2nk_{Cen}D}{T\eta}) - 1) > \log(\det(CA^{-1})\delta^2) \tag{61}
$$

$$
d\log(\frac{2nk_{Cen}D}{T\eta}) - \log(\det(CA^{-1})) - d + 2\log(\frac{1}{\delta}) > 0.
$$

Therefore, we find $\mathcal{G}_{Cen} > 0$. Considering increasing $n$ leads to $nm \geq D$, we derive the upper bound of $\tilde{\mathcal{G}}_{Fed} - \mathcal{G}_{Cen}$ as follows:

$$
\tilde{\mathcal{G}}_{Fed} - \mathcal{G}_{Cen}
$$

$$
\leq \frac{d\log\left(\frac{2n^\gamma k_{Fed}m}{T\eta}\right) - \log\left(\det\left(CA^{-1}\right)\right) + \frac{T\eta}{2n^\gamma k_{Fed}m}\mathrm{tr}\left(CA^{-1}\right) - d + 2\log\left(\frac{1}{\delta}\right) + 2\log\left(nm\right) + 4}{4nm - 2}
$$

$$
- \frac{d\log\left(\frac{2nk_{Cen}D}{T\eta}\right) - \log\left(\det\left(CA^{-1}\right)\right) + \frac{T\eta}{2nk_{Cen}D}\mathrm{tr}\left(CA^{-1}\right) - d + 2\log\left(\frac{1}{\delta}\right) + 2\log\left(D\right) + 4}{4nm - 2}
$$

$$
= \frac{d\log\left(n^{\gamma-1}\right) + \frac{T\eta}{2n^{\gamma+1}k_{Cen}m}\mathrm{tr}\left(CA^{-1}\right) - \frac{T\eta}{2nk_{Cen}D}\mathrm{tr}\left(CA^{-1}\right) + 2\log(nm) - 2\log(D)}{4nm - 2}
$$

$$
= \frac{d\log\left(n^{\gamma-1}\right)}{4nm - 2} + \frac{\frac{T\eta}{2n^{\gamma+1}k_{Cen}m}\mathrm{tr}\left(CA^{-1}\right) - \frac{T\eta}{2nk_{Cen}D}\mathrm{tr}\left(CA^{-1}\right)}{4nm - 2} + \frac{2\log(nm) - 2\log(D)}{4nm - 2} \tag{62}
$$

We separately analyze the values of the three terms in Eq.(62) when $n$ approaches infinity. For the first term, we have

$$
\lim_{n\to\infty} \frac{d\log\left(n^{\gamma-1}\right)}{4nm - 2} = \lim_{n\to\infty} \frac{\frac{(\gamma-1)d}{n}}{4m} = 0. \tag{63}
$$

For the second term, we have

$$
\lim_{n\to\infty} \frac{\frac{T\eta}{2n^{\gamma+1}k_{Cen}m}\mathrm{tr}\left(CA^{-1}\right) - \frac{T\eta}{2nk_{Cen}D}\mathrm{tr}\left(CA^{-1}\right)}{4nm - 2}
$$

$$
= \lim_{n\to\infty} \frac{\frac{T\eta}{2k_{Cen}m}\mathrm{tr}\left(CA^{-1}\right) - \frac{n^\gamma T\eta}{2k_{Cen}D}\mathrm{tr}\left(CA^{-1}\right)}{n^{\gamma+1}\left(4nm - 2\right)} \tag{64}
$$

$$
= \lim_{n\to\infty} \left(\frac{\frac{T\eta}{2k_{Cen}m}\mathrm{tr}\left(CA^{-1}\right)}{n^{\gamma+1}\left(4nm - 2\right)} - \frac{\frac{T\eta}{2k_{Cen}D}\mathrm{tr}\left(CA^{-1}\right)}{4n^2m - 2n}\right) = 0.
$$

For the last term, we derive

$$
\lim_{n\to\infty} \frac{2\log(nm) - 2\log(D)}{4nm - 2}
$$

$$
= \lim_{n\to\infty} \frac{\frac{d}{dn}(2\log(nm) - 2\log(D))}{\frac{d}{dn}(4nm - 2)} \tag{65}
$$

$$
= \lim_{n\to\infty} \frac{\frac{1}{n}}{2m} = 0
$$

By combining Eqs.(63), (64) and (65), we prove

$$
\lim_{n\to\infty} \left(\tilde{\mathcal{G}}_{Fed} - \mathcal{G}_{Cen}\right) \leq 0. \tag{66}
$$

Then, we analyze the case when $m$ tends to positive infinity. Similarly, based on Assumption 5, the upper bound of $\tilde{\mathcal{G}}_{Fed} - \mathcal{G}_{Cen}$ is derived as the following form:

$$\tilde{\mathcal{G}}_{Fed} - \mathcal{G}_{Cen}$$

$$\leq \frac{d \log\left(\frac{n^{\gamma-1}k_{Fed}m}{k_{Cen}D}\right) + \frac{T\eta}{2n^{\gamma+1}k_{Cen}m}\text{tr}\left(CA^{-1}\right) - \frac{T\eta}{2nk_{Cen}D}\text{tr}\left(CA^{-1}\right) + 2\log(nm) - 2\log(D)}{4nm-2}$$

$$= \frac{d\log\left(\frac{n^{\gamma-1}k_{Fed}m}{k_{Cen}D}\right)}{4nm-2} + \frac{\frac{T\eta}{2n^{\gamma+1}k_{Cen}m}\text{tr}\left(CA^{-1}\right) - \frac{T\eta}{2nk_{Cen}D}\text{tr}\left(CA^{-1}\right)}{4nm-2} + \frac{2\log(nm) - 2\log(D)}{4nm-2}. \tag{67}$$

When $m$ approaches infinity, the first term in Eq.(67) becomes:

$$\lim_{m\to\infty} \frac{d\log\left(\frac{n^{\gamma-1}k_{Fed}m}{k_{Cen}D}\right)}{4nm-2}$$

$$= \lim_{m\to\infty} \frac{\frac{d}{dm}\left(d\log\left(\frac{n^{\gamma-1}k_{Fed}m}{k_{Cen}D}\right)\right)}{\frac{d}{dm}(4nm-2)} \tag{68}$$

$$= \lim_{m\to\infty} \frac{\frac{d}{m}}{4n} = 0.$$

The second term becomes:

$$\lim_{m\to\infty} \frac{\frac{T\eta}{2n^{\gamma+1}k_{Cen}m}\text{tr}\left(CA^{-1}\right) - \frac{T\eta}{2nk_{Cen}D}\text{tr}\left(CA^{-1}\right)}{4nm-2}$$

$$= \lim_{m\to\infty} \left(\frac{\frac{T\eta}{2n^{\gamma+1}k_{Cen}}\text{tr}\left(CA^{-1}\right)}{4nm^2-2m} - \frac{\frac{T\eta}{2nk_{Cen}D}\text{tr}\left(CA^{-1}\right)}{4nm-2}\right) = 0. \tag{69}$$

The third term becomes:

$$\lim_{m\to\infty} \frac{2\log(nm) - 2\log(D)}{4nm-2}$$

$$= \lim_{m\to\infty} \frac{\frac{d}{dm}(2\log(nm) - 2\log(D))}{\frac{d}{dm}(4nm-2)} \tag{70}$$

$$= \lim_{m\to\infty} \frac{\frac{1}{m}}{2n} = 0.$$

With Eqs.(68), (69) and (70), we find the below inequality holds for the case of $m$ approaches positive infinity:

$$\lim_{m\to\infty} \left(\tilde{\mathcal{G}}_{Fed} - \mathcal{G}_{Cen}\right) \leq 0. \tag{71}$$

Third, we consider the case when d tends to positive infinity. Here, we denote the model size in the centralized scenario as $\tilde{d}$. Since we attempt to increase the model size $d$ in the federated scenario, we have $d \geq \tilde{d}$. With this condition, there exists a lower bound for the performance gap $\tilde{\mathcal{G}}_{Fed} - \mathcal{G}_{Cen}$, with the following form:

$$\tilde{\mathcal{G}}_{Fed} - \mathcal{G}_{Cen}$$

$$\geq \frac{d\log\left(\frac{2n^{\gamma}k_{Fed}m}{T\eta}\right) - \log\left(\det\left(CA^{-1}\right)\right) + \frac{T\eta}{2n^{\gamma}k_{Fed}m}\text{tr}\left(CA^{-1}\right) - d + 2\log\left(\frac{1}{\delta}\right) + 2\log\left(nm\right) + 4}{4nm-2}$$

$$- \frac{d\log\left(\frac{2nk_{Cen}D}{T\eta}\right) - \log\left(\det\left(CA^{-1}\right)\right) + \frac{T\eta}{2nk_{Cen}D}\text{tr}\left(CA^{-1}\right) - \tilde{d} + 2\log\left(\frac{1}{\delta}\right) + 2\log\left(D\right) + 4}{4D-2}$$

$$= \frac{d\left(\log\left(n^{\gamma-1}\right) - 1\right) + \frac{T\eta}{2n^{\gamma}k_{Fed}m}\text{tr}\left(CA^{-1}\right) - \frac{T\eta}{2nk_{Cen}D}\text{tr}\left(CA^{-1}\right) + \tilde{d}}{4nm-2}$$

$$\tag{72}$$

Based on the Assumption 5, we have

$$n > \sqrt[\gamma-1]{e}$$

$$\log(n) > \frac{1}{\gamma - 1} \tag{73}$$

$$\log\left(n^{\gamma-1}\right) - 1 > 0.$$

Therefore,

$$\lim_{d \to \infty} \left(\tilde{\mathcal{G}}_{Fed} - \mathcal{G}_{Cen}\right)$$

$$\geq \lim_{d \to \infty} \left(\frac{d\left(\log\left(n^{\gamma-1}\right) - 1\right) + \frac{T\eta}{2n^{\gamma+1}k_{Cen}m}\operatorname{tr}\left(CA^{-1}\right) - \frac{T\eta}{2nk_{Cen}D}\operatorname{tr}\left(CA^{-1}\right) + \tilde{d}}{4nm - 2}\right) \tag{74}$$

$$= \infty.$$

Finally, we study the case when $T$ tends to positive infinity. Like the proof for $d$, we represent the number of iterations for the centralized scenario as $\tilde{T}$. Increasing the number of communication rounds $T$ in the federated scenario results in $T \geq \tilde{T}$. Thus, the performance gap $\tilde{\mathcal{G}}_{Fed} - \mathcal{G}_{Cen}$ can be expressed as follows:

$$\tilde{\mathcal{G}}_{Fed} - \mathcal{G}_{Cen}$$

$$= \frac{d\log\left(\frac{2n^\gamma k_{Fed}m}{T\eta}\right) - \log\left(\det\left(CA^{-1}\right)\right) + \frac{T\eta}{2n^\gamma k_{Fed}m}\operatorname{tr}\left(CA^{-1}\right) - d + 2\log\left(\frac{1}{\delta}\right) + 2\log(nm) + 4}{4nm - 2}$$

$$- \frac{d\log\left(\frac{2nk_{Cen}D}{\tilde{T}\eta}\right) - \log\left(\det\left(CA^{-1}\right)\right) + \frac{\tilde{T}\eta}{2nk_{Cen}D}\operatorname{tr}\left(CA^{-1}\right) - d + 2\log\left(\frac{1}{\delta}\right) + 2\log(D) + 4}{4D - 2}$$

$$= \frac{d\log\left(\frac{2n^\gamma k_{Fed}m}{T\eta}\right) + \frac{T\eta}{2n^\gamma k_{Fed}m}\operatorname{tr}\left(CA^{-1}\right) - d\log\left(\frac{2nk_{Cen}D}{\tilde{T}\eta}\right) - \frac{\tilde{T}\eta}{2nk_{Cen}D}\operatorname{tr}\left(CA^{-1}\right)}{4nm - 2}.$$

$$= \frac{d\log\left(\frac{2n^\gamma k_{Fed}m}{T\eta}\right) + \frac{T\eta}{2n^\gamma k_{Fed}m}\operatorname{tr}\left(CA^{-1}\right)}{4nm - 2} - \frac{d\log\left(\frac{2nk_{Cen}D}{\tilde{T}\eta}\right) + \frac{\tilde{T}\eta}{2nk_{Cen}D}\operatorname{tr}\left(CA^{-1}\right)}{4nm - 2} \tag{75}$$

It is easy to recognize that the value of $\tilde{\mathcal{G}}_{Fed} - \mathcal{G}_{Cen}$ depends on the left term in Eq.(75) when $T$ tends to infinity. To understand how this term changes as T increases, we need to compare the impact of $d\log\left(\frac{2n^\gamma k_{Fed}m}{T\eta}\right)$ and $\frac{T\eta}{2n^\gamma k_{Fed}m}\operatorname{tr}\left(CA^{-1}\right)$, which is expressed as follows:

$$\lim_{T \to \infty} \frac{d\log\left(\frac{2n^\gamma k_{Fed}m}{T\eta}\right)}{\frac{T\eta}{2n^\gamma k_{Fed}m}\operatorname{tr}\left(CA^{-1}\right)}$$

$$= \lim_{T \to \infty} \frac{\frac{d}{dT}\left(d\log\left(\frac{2n^\gamma k_{Fed}m}{T\eta}\right)\right)}{\frac{d}{dT}\left(\frac{T\eta}{2n^\gamma k_{Fed}m}\operatorname{tr}\left(CA^{-1}\right)\right)} \tag{76}$$

$$= \lim_{T \to \infty} \frac{-\frac{d}{T}}{\frac{\eta}{2n^\gamma k_{Fed}m}\operatorname{tr}\left(CA^{-1}\right)} = 0.$$

From Eq.(76), we know that

$$\lim_{T \to \infty} \left(d\log\left(\frac{2n^\gamma k_{Fed}m}{T\eta}\right) + \frac{T\eta}{2n^\gamma k_{Fed}m}\operatorname{tr}\left(CA^{-1}\right)\right) = +\infty. \tag{77}$$

In summary, we have

$$\lim_{T \to \infty}\left(\tilde{\mathcal{G}}_{Fed} - \mathcal{G}_{Cen}\right) = \infty. \tag{78}$$

The proof has been completed with the inequalities in Eqs.(66), (71), (74) and (78). $\qquad\square$

## A.2 DETAILED EXPERIMENT SETUP

In this subsection, we present the details of our experiment setup through two tables. Table 1 details the experiment system, covering the specific settings for model architecture, dataset, federated scenario, and training. Table 2 outlines the running environment, including the configuration of the executed codes and the test server.

Table 1: Experiment System Settings.

| System | Value |
|---|---|
| Model Architecture | Vision Transformer (ViT) (Dosovitskiy et al., 2020) |
| | ResNet (He et al., 2016) |
| Dataset | Mini-ImageNet (Vinyals et al., 2016) |
| | CIFAR-10 (Krizhevsky et al., 2009) |
| Range on Communication Rounds | $25 \leq T \leq 100$ |
| Range on Number of Clients | $2 \leq n \leq 100$ |
| Data Distribution on Clients | I.I.D |
| ViT Model Size Options (Millions) | $\{7.91, 15.00, 22.08, 29.17, 36.26,$ |
| | $43.35, 50.44, 57.52, 64.61, 71.70\}$ |
| ResNet Model Size Options (Millions) | $\{4.91, 11.18, 17.45, 23.72, 29.99,$ |
| | $36.26, 42.54, 48.81, 55.08, 61.35\}$ |
| Local Training Epochs | $t = 2$ |
| Batch Size | 256 |
| Base Learning Rate | 1.5e-4 |

Table 2: Running Environment Settings.

| Config | Details |
|---|---|
| Server GPU Count | 8 |
| Server GPU Type | RTX A5000 (24GB) |
| Server CPU Type | AMD EPYC 7513 32-core |
| Programming Language | Python |
| CUDA | 11.3 |
| Framework | PyTorch |

### A.3    ADDITIONAL EXPERIMENTAL RESULTS

Figure 3 shows the results of experiments on testing the strategies of incorporating new clients or adding new data to existing clients. In addition to these two experiments, we have also conducted another two sets of empirical studies on the strategies of scaling up model size or increasing the number of communication rounds, which corresponds to allowing the federated scenario to have an advantage in the parameters $d$ and $T$. Based on the results of the additional experiments demonstrated in Figures 4 and 5, we can recognize that increasing the model size or the number of communication rounds is not able to fully bridge the performance gap between federated and centralized training, which also validates our Theorem 4.

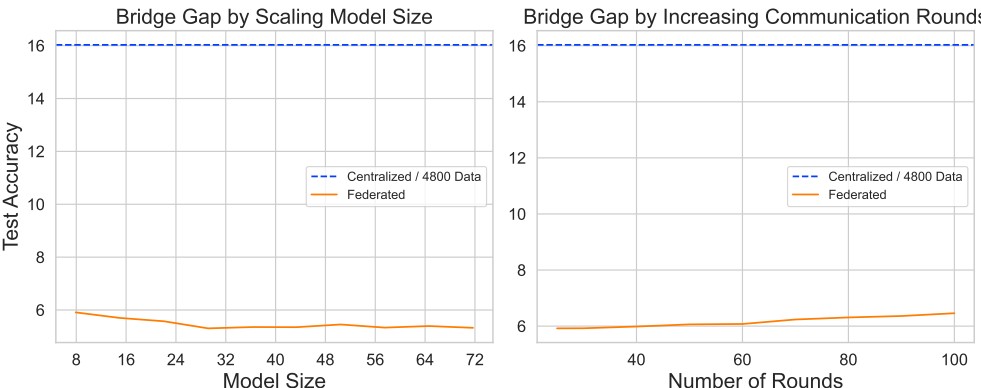

Figure 4: Additional empirical evidence for fully closing the performance gap between federated and centralized training setups. The baseline centralized scenario contains 4800 data, aligned with the centralized scenario in previous experiments. **(Left)** The strategy of scaling model sizes (increasing the model size $d$). **(Right)** The strategy of increasing communication rounds (only increasing the number of communication rounds $T$).

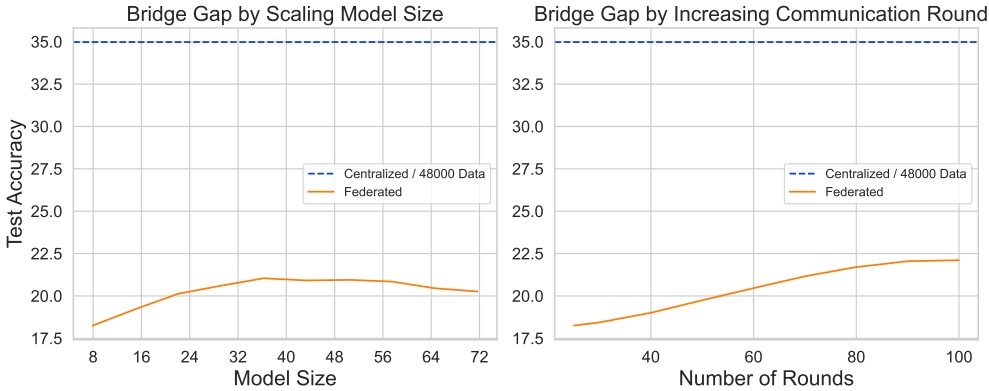

Figure 5: Additional empirical evidence for fully closing the performance gap between federated and centralized training setups. The baseline centralized scenario holds the complete training dataset containing 48000 data. **(Left)** The strategy of scaling model sizes (increasing the model size $d$). **(Right)** The strategy of increasing communication rounds (only increasing the number of communication rounds $T$).

