# OpenReview forum: "Generalization Performance Gap Analysis between Centralized and Federated Learning: How to Bridge this Gap?"
_ICLR.cc/2025/Conference — Submitted to ICLR 2025_

### Official Review · Reviewer_xmQ6 · 2024-10-24

**Soundness:** 2
**Presentation:** 3
**Contribution:** 2
**Rating:** 5
**Confidence:** 4

**Summary:**

This paper theoretically analyzes the generalization performance gap between federated learning and centralized training. It also proves that the complete elimination of the performance gap is only possible by introducing new clients or addind new data.

**Strengths:**

1) This work formulates the generalization performance of federated learning and centralized learning and gives a detailed illustration for the generalization bound.

2) This work proposes a method to close the generalization gap between two learning systems by introducing new clients or new data.

**Weaknesses:**

1) Some related works are missing, e.g., [1] [2] [3]

[1] What Do We Mean by Generalization in Federated Learning? arxiv: 2110.14216, Neurips workshop

[2] Understanding Generalization of Federated Learning via Stability: Heterogeneity Matters AISTATS 2024

[3] Towards Understanding Generalization and Stability Gaps between Centralized and Decentralized Federated Learning arxiv: 2310.03461

2) Strong assumption: The loss function is assumed to be convex and 2-order differentiable, which limits the contributions of the theoretical findings.

3) The method to close the generalization gap seems be to less practical in real-world applications. By introducing new clients or new data, the training data may own different distribution and change the training goal. In this case, it is hard to compare this dynamic system with centralized one.

**Questions:**

1) In Theorem 3, are the lower bound and upper bound necessarily positive?

2) In Line 422, by introducing new clients or new data, will the number of data size mn be increased to be larger than D?

3) In Figure 1, when changing the number of clients, does the total data size on all clients also change? If not, we may not get the conclusion in Section 5.2.1.

---

> ### Author Response · Authors · 2024-11-17
> **Reply to reviewer's comments**
>
> Thanks for reviewing our paper. We appreciate the time and effort you have dedicated to helping us improve our work.  Below are our responses to the points you raised:
> - For Weaknesses:
>     1. Thank you for providing us with these papers. While these studies also explore the generalization of algorithms, their focuses differ from ours. Specifically, $(a)$ the first and second papers aim to understand and improve the generalization ability of federated learning but do not consider comparisons with centralized scenarios; $(b)$ the third paper compares the generalization performance of two different types of federated learning. Moreover, the third paper was published on arXiV after our submission to ICLR, so we were unable to include it in our original submission. In the final version, we will include the three papers in our related work section.
>     2. The convex assumption about the loss function (Assumption 2) is a standard assumption in studies exploring the generalization ability of stochastic algorithms [1, 2]. This assumption makes sense when the loss function is smooth and the stochastic process stabilizes into a low-variance quasi-stationary distribution near a deep local minimum. Its validity has been supported by empirical evidence (see [3, p.1, Figures 1(a) and 1(b), and p.6, Figures 4(a) and 4(b)]). While this assumption simplifies the analysis, it does not fundamentally alter the geometry of the objective function or the resulting generalization ability. General cases can be obtained by translation operations.
>     3. Thank you for raising this important concern. Our theoretical results are based on the IID assumption, ensuring that the proposed strategies are feasible if the new training data is drawn from distributions similar to the existing data.  However, if the new data has significantly different distributions, the situation becomes more complex. While our current analysis does not directly address the non-IID case, we can provide a reasonable hypothesis based on our theoretical framework. Specifically, transitioning from an IID to a non-IID setup would affect Assumption 4 but keep the other setups unchanged, leading to $|A-\bar{A}|^2 \leq \alpha$ and $|\frac{1}{n^\gamma}C - \bar{C}|^2 \leq \beta$, where the hyper-parameters $\alpha \geq 0$ and $\beta \geq 0$ represent the degree of non-IIDness. Under this new assumption, the term $\text{tr}(CA^{-1})$ in Eq.(17) would be reformulated as $\text{tr}(\bar{C}\bar{A}^{-1}) - \text{tr}(CA^{-1}) = \text{tr}(\bar{C}\bar{A}^{-1} - CA^{-1}) \leq \frac{\beta}{n^\gamma}\alpha$. This implies that the generalization gap between the two scenarios increases as the level of non-IIDness grows. Consequently, introducing new training data with distinct distributions in practice would simultaneously decrease the generalization gap (by increasing total data size) and increase it (due to the heightened non-IID level across clients). Since the denominator of Eq.(17) does not include $\text{tr}(CA^{-1})$, our proposed strategies still have the potential to bridge the gap, while the effectiveness would depend on the magnitude of the increase in the degree of non-IIDness across local datasets.
> - (Q1) The lower and upper bounds in Theorem 3 are necessarily positive if Assumption 5 is satisfied. This assumption generally holds in current studies, as state-of-the-art models are typically over-parameterized to achieve superior performance, and federated scenarios often involve a large number of clients.
> - (Q2) By introducing new clients or additional data, the total data size $mn$ in the federated scenario will increase and can exceed the total data size $D$ in the centralized scenario.
> - (Q3) In Figure 1, the experimental results compare the federated and centralized scenarios under equal training conditions. Therefore, the total data size across all clients is kept equal to the total data size in the centralized scenario and remains unchanged throughout the experiments.
>
> [1] Stochastic gradient descent as approximate bayesian inference. Journal of Machine Learning Research, 18(134):1–35, 2017.
>
> [2] Control batch size and learning rate to generalize well: Theoretical and empirical evidence. Advances in neural information processing systems, 32, 2019.
>
> [3] Visualizing the loss landscape of neural nets. Advances in neural information processing systems, 31, 2018.

---

> > ### Comment · Reviewer_xmQ6 · 2024-11-18
> >
> > Thank you for your response. But I would keep my score.
> >
> > My main concern is (from W3 and Q2): when the number of data size mn is increased to be larger than D, is this federated system still comparable with the centralized one? I think this comparison is not fair any more.

---

> > > ### Author Response · Authors · 2024-11-18
> > >
> > > Thank you for your prompt follow-up comment. We would like to further clarify your concern regarding the fairness of the comparison.
> > >
> > > One of the key contributions of our paper is the proof that a generalization gap necessarily exists between federated and centralized learning scenarios with the same amount of training data (i.e., $mn = D$). Theorem 2 provides an analytic solution to this gap, and Theorem 3 establishes non-vacuous lower and upper bounds, rigorously confirming the existence of the generalization gap.
> > >
> > > Building on these findings, we further prove that federated learning must take advantage of certain training parameters to bridge the generalization gap, and the only approach is to add more training data. Specifically, in Theorem 4, assuming that the federated scenario has been allocated with training advantages over the centralized scenario, we prove that this generalization gap is only possible to be completely closed by either introducing new clients (shown as $\lim _{n\to\infty} \tilde{\mathcal{G}} _{Fed} - \mathcal{G} _{Cen} \leq 0$) or adding additional data to existing clients (shown as $ \lim _{m\to\infty} \tilde{\mathcal{G}} _{Fed} - \mathcal{G} _{Cen} \leq 0 $).
> > >
> > > We hope that the above explanation resolves your concern. If you have further questions or require additional details, we are more than happy to provide further explanations. We would appreciate it if you could reconsider the score of our paper.

---

> > > > ### Author Response · Authors · 2024-11-22
> > > > **Look forward to further discussions**
> > > >
> > > > Dear reviewer,
> > > >
> > > > We hope this message finds you well.
> > > >
> > > > Thank you for your earlier follow-up comment. We have provided further clarification on the new question you raised and would like to know if our reply has satisfactorily addressed your concern. If you have further questions or require additional details, we are more than happy to further discuss with you. We would greatly appreciate your reconsideration of the score of our paper.

---

### Official Review · Reviewer_PELM · 2024-10-27

**Soundness:** 3
**Presentation:** 3
**Contribution:** 2
**Rating:** 5
**Confidence:** 3

**Summary:**

This paper primarily investigates the generalization performance gap between centralized and federated learning (FedAvg) algorithms. Previous empirical evidence suggests that the generalization performance in a federated setting is inferior to that in a centralized setting. Based on the Probably Approximately Correct (PAC) framework, this study derives the generalization performance for both centralized and federated paradigms, measures the performance disparity between them, and proposes several methods to bridge this gap, supported by theoretical deductions and validated through experimental evidence.

**Strengths:**

- The theory is solid: This paper introduces the PAC (Probably Approximately Correct) framework and, under certain assumptions, derives the generalization bounds for both centralized and federated settings.
- The conclusions hold significant implications: The results derived from theoretical deductions offer guidance for the practical deployment of Federated Learning (FL) algorithms.

**Weaknesses:**

1. Eq.(7) seems not to capture the situation where a client performs multiple rounds of local updates, such as $K$ rounds, because, in the FedAvg algorithm, clients typically need to update their models locally $K$ times.
2. Assumption 2 appears to be a strong assumption, especially \( F(0) = 0 \), which is often not satisfied in practice. Can some experimental validation be provided?
3. The main focus of this paper is theoretical exploration. The conclusions drawn are generally trivial and do not offer any methodological innovations. What, then, is the technical contribution of this paper?

**Questions:**

please see the weakness.

---

> ### Author Response · Authors · 2024-11-17
> **Reply to reviewer's comments**
>
> Thanks for reviewing our paper. We appreciate the time and effort you have dedicated to helping us improve our work.  Below are our responses to the weaknesses and questions you raised:
> - Our theoretical results are applicable to scenarios where clients perform multiple rounds of local updates, denoted as $t$ in our paper. Here, $t$ can be any positive integer. We consider the impact of $t$ on local SGD updates by including it in the analytic solution of the output hypothesis of federated SGD (Eq.(14)). The theoretical expressions for the generalization bounds (Eqs.(13) and (15)) and the performance gap (Eq.(17)) do not include $t$ as $t$ is simplified through integral operations in the proofs. For further details, please refer to Eqs.(30) and (44) in the Appendix, where this simplification is explained more thoroughly.
> - Assumption 2 is standard on papers studying the generalization ability of stochastic algorithms [1,2], which makes sense when the loss function is smooth and the stochastic process reaches a low-variance quasi-stationary distribution around a deep local minimum. This assumption has been primarily demonstrated by empirical works (see [3, p.1, Figures 1(a) and 1(b) and p.6, Figures 4(a) and 4(b)]). General cases can be obtained by translation operations, which would not change the geometry of the objective function and the corresponding generalization ability.
> - The technical contributions of this paper are clarified as follows:
>     - **A novel perspective on the generalization gap**: Unlike previous works that primarily focus on empirical observations, our paper offers a new theoretical view to analyze the generalization gap. Specifically, we demonstrate that the gap can be formally expressed as the difference between the PAC-Bayes generalization bounds of two scenarios. While PAC-Bayes theory has been extensively studied, this connection between the generalization gap and PAC-Bayes bounds is novel. Furthermore, our derivation reveals how training differences between scenarios lead to the gap, addressing an under-explored question in prior research.
>     - **Understanding the impact of training parameters**: Our derived lower and upper bounds offer valuable evidence for understanding how training parameters influence the generalization gap. Specifically, $(a)$ a gap necessarily exists between centralized and federated scenarios when training conditions are identical; $(b)$ the gap decreases with the number of training rounds $T$ but cannot be entirely eliminated by increasing $T$; $(c)$ the gap increases with the number of clients $n$ and the model size $d$. While previous studies have reported similar trends empirically, they lacked a theoretical foundation to explain why these results occur or whether they hold generally. Our work addresses this gap, providing greater clarity and justifying generality.
>     - **Guidance on bridging the gap**: Beyond identifying the existence and characteristics of the generalization gap, our results offer actionable insights for addressing it. We identify feasible and infeasible strategies for reducing the gap, helping future research focus on promising directions while avoiding ineffective approaches. This guidance enhances the practical relevance of our theoretical contributions.
>
> [1] Stochastic gradient descent as approximate bayesian inference. Journal of Machine Learning Research, 18(134):1–35, 2017.
>
> [2] Control batch size and learning rate to generalize well: Theoretical and empirical evidence. Advances in neural information processing systems, 32, 2019.
>
> [3] Visualizing the loss landscape of neural nets. Advances in neural information processing systems, 31, 2018.

---

> > ### Author Response · Authors · 2024-11-22
> > **Look forward to discuss**
> >
> > Dear reviewer,
> >
> > We hope this message finds you well. We greatly appreciate your time and effort in reviewing our paper.  We have carefully answered all the questions you raised about our paper and look forward to further discussions with you.

---

### Official Review · Reviewer_2KXz · 2024-11-02

**Soundness:** 3
**Presentation:** 3
**Contribution:** 2
**Rating:** 5
**Confidence:** 3

**Summary:**

This work proposes a theoretical understanding of the generalization gap between centralized and federated learning within the PAC-Bayes framework and reveals how to eliminate the performance gap. They also find that advantages in other training resources are not feasible for closing the gap, such as giving larger models or more communication rounds for FL. The empirical results prove the theoretical analysis with different model architectures and datasets.

**Strengths:**

1. This is the first study to understand the generalization performance gap between centralized and federated learning.
2. The paper is well-organized and easy to read.
3. The experiments can prove the theoretical results.

**Weaknesses:**

1. I wonder what are the technical contributions of this paper. It seems that the proposed analysis may appear to be an extension of PAC-Bayes generalization gap to centralized and FL scenarios. It is essential for the authors to underscore their technical contributions in a more prominent manner.
2. Is the analysis for FL based on federated SGD with only one local update (t=1)? Or is the generalization performance gap in this analysis related to the number of local updates?

**Questions:**

See the weakness above.

---

> ### Author Response · Authors · 2024-11-17
> **Reply to reviewer's comments**
>
> Thanks for reviewing our paper. We appreciate the time and effort you have dedicated to helping us improve our work.  Below are our responses to the weaknesses and questions you raised:
> - Thank you for this suggestion. In the final version, we will revise the introduction to more prominently highlight our technical contributions. The key technical contributions are clarified as follows:
>     - **A novel perspective on the generalization gap**: Compared to the previous studies, our work shows a novel perspective to analyze the generalization gap. Specifically, we demonstrate that this gap can be theoretically defined as the difference between the PAC-Bayes generalization bounds of two scenarios. While PAC-Bayes theory has been extensively studied, this connection between the generalization gap and PAC-Bayes bound has not been previously established. Besides, the derivation of this theoretical expression reveals how the generalization gap resulted from the training differences between the two scenarios, filling an under-explored aspect in previous works.
>     - **Understanding the impact of training parameters**: In addition, our derived lower and upper bounds provide a deeper understanding of the factors influencing the generalization gap. Specifically, $(a)$ a gap necessarily exists between the centralized and federated scenarios if two scenarios are provided with equal training conditions; $(b)$ this gap decreases with the number of training rounds $T$ but cannot be completely closed by increasing $T$; $(c)$ this gap increases with the number of clients $n$ and the model size $d$. While previous works have reported similar results, their conclusions were summarized from the experimental results, so they were unable to explain the underlying causes or establish their general applicability. Our work fills this gap by offering a clear explanation and validating the generality.
>     - **Guidance on bridging the gap**: Beyond identifying the existence and properties of the gap, our work provides valuable guidance on strategies for closing it. Specifically, we distinguish between feasible and infeasible approaches for reducing the gap. These insights help future research focus efforts on promising directions, avoiding wasted resources on approaches that are theoretically non-viable.
> - Our theoretical results can apply to different numbers of local updates (denoted by $t$). The theoretical expressions for the generalization bounds (Eqs.(13) and (15)) and the performance gap (Eq.(17)) do not include $t$ because $t$ is simplified through integral operations in the proofs. See Eqs.(30) and (44) in the Appendix for more details.

---

> > ### Author Response · Authors · 2024-11-22
> > **Look forward to discuss**
> >
> > Dear reviewer,
> >
> > We hope this message finds you well. We greatly appreciate your time and effort in reviewing our paper.  We have carefully answered all the questions you raised about our paper and look forward to further discussions with you.

---

> > > ### Comment · Reviewer_2KXz · 2024-11-24
> > > **Question about Technical Contributions**
> > >
> > > Thank you for your response.
> > > For the technical contributions, your analysis presents a novel perspective on the generalization gap, but could you compare your analysis approach to existing PAC-Bayes analyses of centralized and federated learning? This can help to highlight the novel elements in the analysis.

---

> > > > ### Author Response · Authors · 2024-11-25
> > > > **Reply to the question**
> > > >
> > > > Thank you for your follow-up comment. To the best of our knowledge, our paper is the first to compare the generalization between centralized learning (CL) and federated learning (FL) using the PAC-Bayesian theory. Previous works have primarily focused on using PAC-Bayesian generalization bounds to analyze a single learning regime, either CL or FL, without exploring their differences.
> > > >
> > > > Specifically, at earlier stages, PAC-Bayesian theory was predominantly applied to analyze the generalization of CL, focusing on topics such as hyper-parameter tuning, algorithm convergence, and training stability. Examples include:
> > > > - [1] focuses on hyper-parameter tuning for centralized stochastic gradient descent (SGD). It provides a PAC-Bayesian generalization bound for models trained with centralized SGD and shows that a small ratio of batch size to learning rate improves generalization.
> > > > - [2] studies the generalization of stochastic gradient Langevin dynamics (SGLD) for non-convex functions, a special case of SGD with centralized data. They establish $O(\frac{1}{N})$ and $O(\frac{1}{\sqrt{N}})$ generalization bounds using stability and PAC-Bayesian theory, respectively, where $N$ is the total data size.
> > > > - [3] combines PAC-Bayes with algorithmic stability to derive a generalization bound for centralized SGD, applicable to all posterior distributions. Their work also introduces a sampling strategy that adapts to training data for improved performance.
> > > >
> > > > Recently, with the rising interest in FL, researchers have started applying the PAC-Bayesian generalization bound to address challenges unique to FL, such as non-IID local data, personalization, and the influence of training parameters on generalization. For example,
> > > > - [4] addresses non-IID challenges in FL, proposing a non-vacuous PAC-Bayes generalization bound for non-IID local data. They define a novel FL objective that enhances training efficiency while preserving the privacy of clients.
> > > > - [5] formulates personalized FL (PFL) within a PAC-Bayesian framework. They propose an algorithm that adapts the global model to individual clients by minimizing the generalization bound on the average true risk across clients.
> > > > - [6] derives a PAC-Bayes generalization bound for federated stochastic algorithms to study the impact of communication rounds on generalization error. They prove that excessive communication reduces generalization performance, and the generalization error of FL decreases faster than that of CL by a factor of $O(\sqrt{\log(n)/n}$, where $n$ is the number of participating clients.
> > > >
> > > > Among the above works,  $[1]$ and $[6]$ are most closely related to our work, as they both derive a PAC-Bayesian generalization bound and analyze the effects of training parameters on generalization in CL and FL respectively. However, they did not consider the generalization gap between CL and FL. Our analysis approach is novel in establishing the generalization bounds for both CL and FL under equal training conditions and defining the generalization gap as the difference between the two bounds. According to the established gap, we identify not only the impact of the number of communication rounds $T$ on this gap but also the effects of other parameters that do not appear in $[1]$ and $[6]$, such as the number of clients $n$ and the model size $d$. Furthermore, we extend our theoretical analysis to discover the feasible training advantages that can be assigned to FL to completely close the generalization gap. Our results directly reveal whether the four parameters ($n$, $d$, the average local date size $m$, and $T$) have the potential to bridge the gap, whereas the prior studies have only offered vague insights on this matter. For instance, the results in $[6]$ can indirectly reflect that having an advantage on $T$ is likely to be infeasible as over-increasing $T$ degrades FL generalization.
> > > >
> > > > [1] Control batch size and learning rate to generalize well: Theoretical and empirical evidence. Advances in neural information processing systems, 32, 2019.
> > > >
> > > > [2] Generalization bounds of sgld for non-convex learning: Two theoretical viewpoints. In Conference on Learning Theory, pp. 605–638. PMLR, 2018.
> > > >
> > > > [3] A pac-bayesian analysis of randomized learning with application to stochastic gradient descent. Advances in Neural Information Processing Systems, 30, 2017.
> > > >
> > > > [4] Federated pac-bayesian learning on non-iid data. In ICASSP 2024-2024 IEEE International Conference on Acoustics, Speech and Signal Processing (ICASSP), pp. 5945–5949. IEEE, 2024.
> > > >
> > > > [5] Personalized federated learning of probabilistic models: A pac-bayesian approach. arXiv preprint arXiv:2401.08351, 2024.
> > > >
> > > > [6] Lessons from generalization error analysis of federated learning: You may communicate less often! In Forty-first International Conference on Machine Learning, 2024.

---

### Official Review · Reviewer_nwfw · 2024-11-03

**Soundness:** 2
**Presentation:** 2
**Contribution:** 1
**Rating:** 3
**Confidence:** 4

**Summary:**

This paper studies the generalization performance of federated learning (FL) under local SGD updates, aiming at understanding why the generalization of federated learning is worse than that of centralized learning observed by empirical results. The authors provide theoretical results to show that the generalization performance gap between federated learning and its centralized counterpart given the same training resources is unavoidable. Based on their theory, the authors further claim that the performance gap can be possibly eliminated when introducing new clients or adding data to existing clients.

**Strengths:**

Both generalization bounds for FedAvg and centralized minibatch SGD are provided. Then based on derived generalization bounds for two scenarios, the analytic expression of the performance gap is derived, which shows that such gap is unavoidable under the same training resources. Two strategies, including introducing a new client or adding more data to existing clients, are provided to bridge the performance gap, which are supported by theoretical justifications.

**Weaknesses:**

1. The main weakness of the paper lies in Theorem 3. Actually, the metric of defined performance gap (i.e., $G_{Fed} - G_{Cen}$) seems unreasonable. Firstly, these are only upper bounds of the true generalization gaps, which are probably too conservative to reflect true generalization performances of two scenarios. Secondly, even the generalization gap (i.e., $R(Q) - \hat{R}(Q)$) is not a good metric to study the performance gap. Note that such generalization gap measures the difference between test loss and training loss. It is practically possible that the test losses for both cases are comparable while the training losses remain a large gap, especially when the training rounds of two cases are not equal (i.e., in the paper the rounds for FL is $n$ times less than centralized case). A more reasonable metric to quantify the performance gap should be $R(Q_{Fed}) - R(Q_{Cen})$.

2. In Theorem 3, the bounds does not reflect the dependence on the number of local updates (i.e., $t$). Recall that when $t=1$, FedAvg is exactly minibatch SGD, which thus should have the same generalization performance. However, the provided bounds does not capture this fact.

3. The assumptions are quite strong, especially Assumptions 2 and 3. These assumptions are hard to satisfy for modern neural networks.

4. In experiments, the learning rates for FL and centralized learning are set identical, which might not be reasonable. Note that for FL the learning rate is usually scaled inversely w.r.t. number of local updates $t$. Then for the i.i.d. case, the "effective" updates of FedAvg is actually $t T/ n$, compared with minibatch SGD. Then with identical learning rates, FL may suffer from worse generalization as shown in (Hardt et al., 2016). Moreover, non-i.i.d. case for FL is not considered.

Reference:
Hardt, M., Recht, B., & Singer, Y. (2016, June). Train faster, generalize better: Stability of stochastic gradient descent. In International conference on machine learning (pp. 1225-1234). PMLR.

**Questions:**

1. Could the authors explain why the difference of generalization upper bounds is a good and reasonable metric to evaluate the performance gap? As in Weakness 1, why not use the difference between test losses?

2. Why do not the bounds in Theorem 3 vanish when $t=1$, as in this case FedAvg and minibatch SGD are the same?

3. What the experimental results of the non-i.i.d. case for FL? How does it compare to the centralized setting? Why are the learning rates for both settings set to be the same?

4. The assumptions are too restrictive. Could them be released, e.g., convex but not quadratic or even non-convex losses?

5. In Section 5.2.2, are the total numbers of data points in both settings the same?

6. In the proof, it seems that SGD is treated as its continuous limit. However, there is some difference between the analyses in continuous and discrete settings. Do the theoretical conclusions still hold in the discrete case? Why not directly analyze SGD?

---

> ### Author Response · Authors · 2024-11-17
> **Reply to reviewer's comments**
>
> Thanks for reviewing our paper. We appreciate the time and effort you have dedicated to helping us improve our work.  Below are our responses to the weaknesses and questions you raised:
> - (Q1 and W1) Compared to the training dataset, the test dataset is often unknown and subject to much greater uncertainty. Without knowing the latent distribution of the test data, the test loss $R$ is not available and cannot be directly used to evaluate the generalization ability of an algorithm. Therefore, it is common practice to use the available training loss $\hat{R}$ to estimate $R$. A well-generalizing algorithm will have a small difference between $\hat{R}$ and $R$, termed the generalization error. This approach is standard in studies on the generalization ability of stochastic algorithms [1,2], and we adopt this definition in our paper. Furthermore, the lower bound indicates the possible minimum generalization error, corresponding to the outcome under ideal conditions. However, due to the uncertainty in the test dataset, such ideal outcomes are rarely achieved. In contrast, the upper bound provides a conservative and robust estimate of the generalization performance, making it more practical for managing the risk of inaccurate estimation.
> - (Q2 and W2) Our theoretical analysis supports any positive integer $t$ for the number of local updates. The generalization bounds (Eqs.(13) and (15)) and the performance gap (Eq.(17)) do not explicitly include $t$ because it is simplified through integration during the proof. Detailed explanations can be found in the Appendix (see Eqs.(30) and (44)). Moreover, federated SGD and centralized SGD differ significantly even when $t=1$. Federated SGD operates on decentralized data and involves local SGD updates and aggregation of local models in each iteration. These distinctions lead to different training outcomes between the two approaches, making the bound of the generalization gap not vanish.
> - (Q3 and W4)
>     - Our theoretical analysis and experiments do not examine the non-IID case as we adopt the IID setup to simplify the comparison problem. We plan on studying the impact of non-IID data on the generalization gap in future work. Based on our current theoretical analysis, we guess that our theoretical results may extend to the non-IID case with some modifications. Specifically, a non-IID setup would affect Assumption 4 while leaving other setups unchanged, resulting in $|A-\bar{A}|^2 \leq \alpha; |\frac{1}{n^\gamma}C - \bar{C}|^2 \leq \beta$ where $\alpha \geq 0$ and $\beta \geq 0$ quantify the degree of non-IIDness. Following this new assumption, the term $\text{tr}(CA^{-1})$ in Eq.(17) will be re-formulated as $(\text{tr}(\bar{C}\bar{A}^{-1}) - \text{tr}(CA^{-1})) = \text{tr}(\bar{C}\bar{A}^{-1} - CA^{-1}) \leq \frac{\beta}{n^{\gamma}}\alpha$, implying that the generalization gap between two scenarios increases with the non-IID level. However, as the denominator of Eq.(17) does not include $\text{tr}(CA^{-1})$, we can recognize that the proposed two strategies that increase the total training data in federated scenarios still have the potential to bridge the gap.
>     - We agree that the learning rate affects the generalization ability of an algorithm, as supported by prior work [2]. However, our experiments are designed to validate the correctness of our theoretical results. To align with the problem setup in our theoretical analysis, we use identical learning rates for both centralized and federated scenarios. Additionally, adjusting the learning rate of federated learning to its optimal value would represent a best-case scenario for the generalization gap. Since we express the gap by the difference between the upper bounds of the generalization error, utilizing identical learning rates is more suitable, which corresponds to one of the worst-case scenarios.

---

> ### Author Response · Authors · 2024-11-17
>
> - (Q4 and W3) Assumptions 2 and 3 are commonly used in studying the generalization of stochastic algorithms [1,2]. Previous papers have found that the stochastic process can be modeled as the multivariate Ornstein-Uhlenbeck process [3] consisting of three regimes. At the beginning, there is a search phase where the algorithm approaches the optimum. In this early phase, our assumptions may not hold. However, with more iterations, there is a second phase where SGD has converged to the vicinity of a local minimum. Here, the objective looks already quadratic if the loss function is smooth, but the gradient noise is small relative to the average gradient. Thus SGD takes a relatively directed path towards the optimum. This is the regime where Assumption 2 becomes valid. A recent paper has also validated this with experimental results (see [4, p.1, Figures 1(a) and 1(b) and p.6, Figures 4(a) and 4(b)]). Finally, after a sufficient number of iterations, the iterates of SGD are near the local optimum, where the average gradient is small, and the sampling noise becomes more important. In this final phase, the iterates can be approximated as samples from the stationary distribution. Since this phase follows the second phase, the covariance $\Sigma$ of the stationary distribution is closely linked to the Hessian matrix $A$ at the optimum. Assumption 3 provides a reasonable estimation of this connection.
> - (Q5) In the experiment in Section 5.2.2, the federated scenario has an advantage over the centralized scenario in one training parameter. If the advantage is in the number of clients $n$ or the average amount of local data $m$, the total data will differ between the federated and centralized scenarios. If the advantage is in the model size $d$ or the number of training rounds $T$, the total data will remain equal in both scenarios.
> - (Q6)
>     - Analyzing federated SGD in a discrete setting introduces a summation symbol ($\sum_{0}^t$) over the number of local updates $t$, which is difficult to simplify in the derivation. In contrast, by treating the iterates of federated SGD as its continuous limit, we can simplify the analytic solution for the SGD iterates using integral operation. Besides, to estimate true generalization performance, our analysis assumes that the models are well-trained, which typically involves a large number of iterations. When $t$ approaches a large value, it is only feasible to derive the analytic solution in a continuous setting.
>     - Results from the continuous analysis can often approximate those from discrete settings when derived through reasonable discretization (e.g., partitioning the continuous domain into sufficiently small grids). In our problem, each grid is an iterate of the SGD update. Since the training will be repeated for a significant number of iterations to ensure that the model is well-trained, individual grids represent only an extremely small fraction of the entire optimization domain. Therefore, our theoretical conclusions derived under a continuous setting also hold approximately in the discrete case.
>
> [1] Stochastic gradient descent as approximate bayesian inference. Journal of Machine Learning Research, 18(134):1–35, 2017.
>
> [2] Control batch size and learning rate to generalize well: Theoretical and empirical evidence. Advances in neural information processing systems, 32, 2019.
>
> [3] On the theory of the brownian motion. Physical review, 36(5):823, 1930.
>
> [4]  Visualizing the loss landscape of neural nets. Advances in neural information processing systems, 31, 2018.

---

> > ### Author Response · Authors · 2024-11-22
> > **Look forward to discuss**
> >
> > Dear reviewer,
> >
> > We hope this message finds you well. We greatly appreciate your time and effort in reviewing our paper.  We have carefully answered all the questions you raised about our paper and look forward to further discussions with you.

---

> > ### Comment · Reviewer_nwfw · 2024-11-26
> >
> > I do not think my concerns are addressed in the sense of the following:
> >
> > 1. For W1 and Q1, my main concern is that the generalization bound is not a good metric to compare SGD and FedAvg, due to that the bound can be too conservative to capture the real performance of the algorithm.
> >
> > 2. For W2 and Q2, could authors explain why FedAvg doe not reduce to SGD when $t=1$? If I understand correctly, when $t=1$ there is only one time of local update for each client, which implies the server's model follows the update of centralized SGD. To be specific, note that $\theta_i(j+1) = \bar{\theta}(t) - \eta \nabla F(\bar{\theta}(t);\zeta_i)$ and hence $\bar{\theta}(t+1) = \frac{1}{n}\sum_{i=1}^n \theta_i(j+1) = \bar{\theta}(t) - \eta \frac{1}{n}\sum_{i=1}^n \nabla F(\bar{\theta}(t);\zeta_i)$, corresponding to centralized SGD.
> >
> > 3. For Q3, the authors said they used identical learning rates in experiments in order to align with their theory. However, the authors acknowledged that identical learning rates are not fair to compare centralized SGD and FedAvg given i.i.d. data. Then, does it mean the theory should also be modified to ensure a fair comparison?
> >
> > 4. For Q5, I agree with Reviewer xmQ6 that the comparison is unfair if the total number of data is not equal. Theoretically, it is well-known that the generalization error scales with the order $O(1/\sqrt{N})$, where $N$ is the total number of data. Thus, I am not convinced that the experiment truly reflects the theory.
> >
> > 5. For Q6, I know that the continuous case can be an approximate of the discrete counterpart by discretization. However, I am concerned that the approximation error can be accumulated step by step, which make the final solution totally different. It would be good to analyze the discrete version directly.

---

> > > ### Author Response · Authors · 2024-11-29
> > >
> > > Thanks for your follow-up comment. We would like to provide further clarifications to address the concerns you have mentioned about our paper.
> > >
> > > 1. As stated in our previous reply, estimating the true expected risk is generally impractical because the latent distribution of the test dataset is unknown, unlike the training dataset. Therefore, it is widely accepted in early [2,5,6] and recent literature [7,8,9] that the generalization performance should be estimated based on the empirical risk over the training dataset, with the generalization bound serving as a standard measure of generalization ability. **To the best of our knowledge, there is no other better metric to describe the generalization difference between two different training scenarios.** We would appreciate it if the reviewer could suggest a better metric for our further study. Furthermore, the generalization bounds and the generalization gap shown in our paper are also non-trivial. The generalization bounds for each training scenario are rigorously established through the classical PAC-Bayes framework (Lemma 1). On the other hand, we have derived NON-VACUOUS lower and upper bounds for the generalization gap using the monotonicity between the gap and the number of clients (Theorem 3). The finding of these two bounds supports the correctness of our defined generalization gap.
> > > 2. **Federated SGD does not reduce to centralized SGD with $t=1$ due to differences in training iterations and data distribution.** These distinctions form the theoretical foundation of our analysis. Below we refer to our paper and highlight the mathematical formulations for federated and centralized SGD to help understand their differences. In federated scenarios with the number of local updates being set to $t=1$, the update to the global model $\bar{\theta}_{i}$ at each iteration can be written as follows:
> > >
> > > $$
> > > \begin{aligned}
> > >         \bar{\theta} _{i}(j+1) &= \frac{1}{n} \sum\limits _{i = 1}^n \theta _{i}(j+1) \quad (\text{model aggregation (Eq.(6))}) \\\\
> > >         &= \frac{1}{n} \sum\limits _{i = 1}^n (\bar{\theta} _{i}(j) - \eta \nabla _{\bar{\theta} _{i}(j)}\mathbb{E} _{\zeta_i \sim \mathcal{D} _i} F(\bar{\theta} _i(j); \zeta _i)) \quad (\text{local SGD update (Eq.(7))}) \\\\
> > >         &= \frac{1}{n} \sum\limits _{i = 1}^n \bar{\theta} _i(j) - \eta \frac{1}{n} \sum\limits _{i = 1}^n \frac{1}{S} \sum\limits _{s\in \mathcal{S}}  \nabla _{\bar{\theta} _{i}(j)}\mathbb{E} _{\zeta _i \sim \mathcal{D} _i} F _s(\bar{\theta} _i(j); \zeta _i),
> > > \end{aligned}
> > > $$
> > >
> > > where $j$ is the round index, $\mathcal{D}_i$ is the local dataset on client $i$ and $S$ is the batch size. In contrast, the SGD iterate in a centralized setup with $t=1$ is formulated as below:
> > >
> > > $$
> > > \begin{aligned}
> > >             \theta _{Cen}(j+1) &= \theta _{Cen}(j) - \eta \frac{1}{S} \sum\limits _{s \in \mathcal{S}} \nabla _{\theta _{Cen}(j)} \mathbb{E} _{\zeta \sim \mathcal{D}} F _s(\theta _{Cen}(j); \zeta)), \quad (\text{refers to Eq.(5)})
> > >         \end{aligned}
> > > $$
> > >
> > > where $\mathcal{D}=\bigcup_{i=1}^n \mathcal{D}_i$ is the global dataset. According to these two equations, we can discover that the SGD iteration significantly differs between federated and centralized setups. Besides, distinct data distributions also affect the iteration, further reinforcing the differences in results.
> > >
> > > 3. We have carefully checked this reply and found that the reviewer may have misunderstood our previous response. We did not acknowledge that identical learning rates are not fair for the comparison. For a fair comparison between federated and centralized scenarios, the model needs to remain consistent, and training iterations should also be controlled to be as similar as possible. Since we agree with the previous work [2] that learning rates affect the generalization of stochastic algorithms, we try to limit the training differences sorely to the scenario distinction between federated and centralized learning. **As such, identical learning rates and batch sizes are used for both scenarios to ensure fairness.** Using different learning rates would instead violate this principle and make the comparison unfair.

---

> ### Author Response · Authors · 2024-11-29
>
> 4. **The experiments in Section 5.2.2 are not intended to compare the generalization between centralized and federated SGD under equal training conditions.** Our theoretical analysis demonstrates that a generalization gap necessarily exists between the two scenarios when training resources, including the total data size, are equal (see Theorem 3). Therefore, the complete close of this gap would require the federated scenario to have advantages in training resources (see Theorem 4). To empirically validate which advantage can bridge the gap, we conduct experiments to compare a centralized scenario with a federated scenario that has an advantage in one training resource (i.e. the number of clients $n$, the average data across clients $m$, the model size $d$ and the communication rounds $T$). The experimental results are provided in Section 5.2.2 and Appendix A.3. Based on these empirical results, Theorem 4 is validated, showing that the gap can only be fully closed by introducing new clients or adding data to existing clients.
> 5. **We would like to emphasize that the approximation is valid, and the concern of error accumulation does not apply in our case.** In our theory, expressing the SGD iteration in discrete settings primarily impacts the analytic solution. Specifically, Eq.(14) would be changed to:
> $$
> \theta _{Fed}(T) = \theta _i(0) e^{-T\bar{A}t} + T \sqrt{\frac{\eta}{k _{Fed}m}} \sum^{t}  e^{-T\bar{A}(t-t')} \bar{B} dW(t').
> $$
> Therefore, the difference between the continuous and discrete formulations is captured by the approximation error between Eq.(14) and the above equation, which does not involve further error accumulation. Since each grid represents a single SGD iterate and training typically involves numerous iterations to ensure the model is well-trained, an individual grid contributes only a tiny fraction of the entire optimization domain. As a result, results from the continuous analysis closely align with those from a discrete analysis. Furthermore, our paper focuses on analyzing the generalization gap between scenarios, not deriving precise analytic solutions for SGD iterates. This approximation does not lead to a wrong understanding of the generalization bound and the generalization gap. Continuous approximations are widely used for studying the generalization of stochastic algorithms, as shown in previous works [1,2], which also validate their theoretical conclusions through extensive experiments.
>
> [1] Stochastic gradient descent as approximate bayesian inference. Journal of Machine Learning Research, 18(134):1–35, 2017.
>
> [2] Control batch size and learning rate to generalize well: Theoretical and empirical evidence. Advances in neural information processing systems, 32, 2019.
>
> [5] Generalization bounds of sgld for non-convex learning: Two theoretical viewpoints. In Conference on Learning Theory, pp. 605–638. PMLR, 2018.
>
> [6] A pac-bayesian analysis of randomized learning with application to stochastic gradient descent. Advances in Neural Information Processing Systems, 30, 2017.
>
> [7] Federated pac-bayesian learning on non-iid data. In ICASSP 2024-2024 IEEE International Conference on Acoustics, Speech and Signal Processing (ICASSP), pp. 5945–5949. IEEE, 2024.
>
> [8] Personalized federated learning of probabilistic models: A pac-bayesian approach. arXiv preprint arXiv:2401.08351, 2024.
>
> [9] Lessons from generalization error analysis of federated learning: You may communicate less often! In Forty-first International Conference on Machine Learning, 2024

---

> ### Comment · Reviewer_nwfw · 2024-12-02
>
> Thank authors for the clarification, but my concerns still remain unresolved. Thus, I will keep my score.
>
> When $t=1$, even in the heterogeneous setting, centralized SGD and FedAvg are statistically the same. To see this, recall the fact $\frac{1}{S} \sum_i E_{\zeta_i \sim D_i}(\cdot) = E_{\xi \sim D} (\cdot)$ due to independence across $D_i \forall i$ and its underlying distribution.

---

> ### Author Response · Authors · 2024-12-03
>
> Thank you for your reply before the rebuttal deadline.
>
> Clarification of the differences between centralized and federated SGD when $t=1$ has been provided in the second point of our previous response. We are surprised that the reviewer still insists that centralized SGD and FedAvg are statistically the same. If they are the same, the generalization performance of FL and CL should be the same (statistically). It is exactly our paper's contribution to prove that the gap necessarily exists between the generalization performance of FL and CL, and we further derive the non-vacuous bounds for this gap. In addition, many previous papers have empirically revealed this performance gap. For example, in paper [10], Table 1 compares the Top-1 accuracies between their proposed method FedU on a federated setup with each client conducting $E = 1$ local epochs and a centralized baseline using BYOL [11]. The accuracy of FedU is $86.48 \\%$, while the accuracy of centralized BYOL is $91.85 \\%$. This performance gap of nearly $5\\%$ provides clear empirical evidence that training differences inherently exist between the two setups.
>
> To help the reviewer better understand our work, we would like to further explain the underlying cause of the performance gap between centralized SGD and FedAvg. The main reason lies in the differences on the training datasets and the training processes. According to the equation $\frac{1}{S} \sum _i \mathbb{E} _{\zeta _i \sim \mathcal{D} _i} = \mathbb{E} _{\xi \sim \mathcal{D} _i}$ provided in the reviewer's reply, it seems the reviewer assumes that the average of the local SGD iterates directly equals the global SGD iterates. However, this equation does not hold, as the local dataset on each client is only a subset of the global dataset. Even though the data sets have the same statistical distribution, the descent direction of the gradients calculated from local SGD is not as same as the direction of the gradients calculated from global SGD (see Figure 3 in [10]). If the federated scenario is constructed in a non-i.i.d setup with highly heterogeneous local data, then the divergence on updating directions will be even greater. A recent paper accepted in ICML has also claimed that the divergence issue cannot be simply resolved through the averaging operation used in FedAvg and theoretically studied this inequality (see Eq.(3) in [12]).
>
> Again, we refer to our paper and emphasize the significant differences between the training processes of federated and centralized SGD. Specifically, the iterates of federated SGD with $t=1$ are given by:
> $$
> \bar{\theta} _{i}(j+1) = \frac{1}{n} \sum\limits _{i = 1}^n \bar{\theta} _i(j) - \eta \frac{1}{n} \sum\limits _{i = 1}^n \frac{1}{S} \sum\limits _{s\in \mathcal{S}}  \nabla _{\bar{\theta} _{i}(j)}\mathbb{E} _{\zeta _i \sim \mathcal{D} _i} F _s(\bar{\theta} _i(j); \zeta _i), \quad (1)
> $$
> whereas the iterates of centralized SGD with $t=1$ are written as:
> $$
>  \theta _{Cen}(j+1) = \theta _ {Cen}(j) -\eta \frac{1}{S} \sum\limits _{s\in \mathcal{S}} \nabla _{\theta _{Cen}(j)} \mathbb{E} _{\zeta \sim \mathcal{D}} F_s(\theta _{Cen}(j); \zeta)). \quad (2)
> $$
> By comparing Eq.(1) and Eq.(2) provided above, we can find that the left term in the right part of Eq.(1) is different from that in Eq.(2), with an additional averaging operation in Eq.(1). Moreover, the right term in the right part of Eq.(1) and Eq.(2) also differs to each other, with not only an extra averaging operation across clients but also the data sampled from different distributions.
>
> We hope that the above explanation resolves your concern. If you have further questions or require additional details, we are more than happy to provide further clarifications. We would appreciate it if you could consider increasing your score about our paper.
>
>
> [10] Collaborative unsupervised visual representation learning from decentralized data. In Proceedings of the IEEE/CVF international conference on computer vision, pp. 4912–4921, 2021.
>
> [11] Bootstrap your own latent - a new approach to self-supervised learning. In Advances in Neural Information Processing Systems, volume 33, pages 21271–21284, 2020.
>
> [12] FedSC: Provable Federated Self-supervised Learning with Spectral Contrastive Objective over Non-iid Data. In Forty-first International Conference on Machine Learning.

---

### Author Response · Authors · 2024-11-28
**Upload a revised version**

Dear reviewers,

Thank you for your valuable feedback, which has greatly helped us improve our paper. Based on your comments, we have uploaded a revised version to address the points you raised. The main changes between the new version and the original one are as follows:

- Revised the summary of our contributions in introduction to highlight the novelty of our analysis.
- Included the three papers suggested by the reviewer xmQ6 in related works.
- Provided a clearer justification for Assumption 2 to help understand why it can be accepted.
- Added a clarification in the theoretical analysis to show that the number of local SGD updates $t$ can be any positive integer and explain why $t$ is not explicitly included in the analytic solutions.
- Improved the readability of the assumptions that we have made by separating their justifications from the main content.
- Made the conclusion more concise to comply with the page limitation.

---

### Author Response · Authors · 2024-12-01
**Follow-up Discussions and Final Considerations**

Dear reviewers,

Thank you for taking the time to thoroughly review our paper and for engaging in thoughtful discussions with us over the past two weeks. We greatly appreciate your valuable comments, which have helped us refine and improve our work.

As the rebuttal deadline is approaching, we kindly ask that if you have any remaining questions or concerns regarding our paper, please let us know at your earliest convenience so that we can address them and get back to you before the deadline.

If our reply has resolved your concerns, we sincerely hope that you consider raising your rating of our paper. We would be deeply grateful for your recognition.

Thank you once again for your time and kindness.

---

### Meta-Review · Area_Chair_bq4s · 2024-12-15

**Metareview:**

Thank you for your submission and your thoughtful responses to reviewer feedback. While I acknowledge that some of the criticisms raised during the review process were not fully justified and that the authors have addressed many concerns in their rebuttal, I must recommend rejecting this paper for the following reasons:

1. **Lack of Reviewer Support**: None of the reviewers strongly supported the paper for acceptance. Most highlighted significant issues with the clarity of contributions, methodological novelty, and the applicability of theoretical assumptions.

2. **Presentation and Contributions**: Despite the authors' revisions, the exact contributions of this paper remain difficult to discern. Reviewers expressed concerns that the work does not sufficiently emphasize its technical innovation or situate itself clearly within the broader landscape of related research.

I encourage the authors to address these concerns and resubmit to a future venue after substantial revision.

**Additional Comments On Reviewer Discussion:**

Consensus of all the reviewers to recommend rejection.

---

### Decision · Program_Chairs · 2025-01-22

Reject